

# Long-term chemical analysis and organic aerosol source apportionment at 9 sites in Central Europe: Source identification and uncertainty assessment

Kaspar R. Daellenbach[1], Giulia Stefenelli[1], Carlo Bozzetti[1], Athanasia Vlachou[1], Paola Fermo[2], Raquel Gonzalez[2], Andrea Piazzalunga[3,*], Cristina Colombi[4], Francesco Canonaco[1], Christoph Hueglin[5], Anne Kasper-Giebl[6], Jean-Luc Jaffrezo[7], Federico Bianchi[1,**], Jay G. Slowik[1], Urs Baltensperger[1], Imad El-Haddad[1], and André S. H. Prévôt[1]

[1]Paul Scherrer Institute (PSI), 5232 Villigen-PSI, Switzerland
[2]Università degli Studi di Milano, 20133 Milano, Italy
[3]Università degli Studi di Milano-Bicocca, 20126 Milano, Italy
[4]ARPA Lombardia, Regional Centre for Air Quality Monitoring, 20122 Milan, Italy
[5] Swiss Federal Laboratories for Materials Science and Technology, 8600 Dübendorf, Switzerland.
[6] Institute of Chemical Technologies and Analytics, Vienna University of Technology, 1060 Wien, Austria
[7]Université Grenoble Alpes, CNRS, IGE, 38000 Grenoble, France
[*]now at: Water and Soil Lab, 24060 Entratico, Italy
[**]now at: Department of Physics, University of Helsinki, 00014 Helsinki, Finland

Correspondence to: André S. H. Prévôt (andre.prevot@psi.ch) and Imad El Haddad (imad.el-haddad@psi.ch)

**Abstract.** Long-term monitoring of the organic aerosol is important for epidemiological studies, validation of atmospheric models, and air quality management. In this study, we apply a recently developed filter-based offline methodology of the aerosol mass spectrometer to investigate the regional and seasonal differences of contributing organic aerosol sources. We present offline-AMS measurements for particulate matter smaller than 10 µm 9 stations in central Europe with different exposure characteristics for the entire year of 2013 (819 samples). The focus of this study is a detailed source apportionment analysis (using PMF) including in-depth assessment of the related uncertainties. Primary organic aerosol (POA) is separated in three components: hydrocarbon-like OA which is related to traffic emissions (HOA), cooking OA (COA), and biomass-burning OA (BBOA). We observe enhanced production of secondary organic aerosol (SOA) in summer, following the increase in biogenic emissions with temperature (summer oxygenated OA, SOOA). In addition, a SOA component was extracted that correlated with anthropogenic secondary inorganic species which is dominant in winter (winter oxygenated OA, WOOA). A factor (SC-OA) explaining sulfur-containing fragments ($CH_3SO_2^+$), which has an event-driven temporal behavior, was also identified. The relative yearly average factor contributions range for HOA from 3 to 15%, for COA from 3 to 31%, for BBOA from 11 to 61%, for SC-OA from 5 to 23%, for WOOA from 14 to 28%, and for SOOA from 14 to 40%. The uncertainty of the relative average factor contribution lies between 5 and 9% of OA. At the sites north of the alpine crest, the sum of HOA, COA, and BBOA (POA) contributes less to OA (POA/OA=0.3) than at the southern alpine valley sites (0.6). BBOA is the main contributor to POA with 88% in alpine valleys and 43% north of the alpine crest. Furthermore,



the influence of primary biological particles (PBOA), not resolved by PMF, is estimated and could contribute significantly to OA in PM10.

## 1    Introduction

The development and field deployment of the Aerodyne aerosol mass spectrometer (AMS, Canagaratna et al., 2007) have greatly improved air quality monitoring by providing real-time measurements of the non-refractory (NR) submicron aerosol (PM1) components. The application of factor analysis on the collected organic aerosol (OA) mass spectra enabled the efficient disentanglement of aerosol factors, which could be subsequently related to specific aerosol sources and processes (Lanz et al., 2007, 2010; Jimenez et al., 2009; Ulbrich et al., 2009, Zhang et al., 2011; Ng et al., 2010; Crippa et al., 2014). Factors typically extracted include directly emitted primary OA (POA) from biomass burning (BBOA) or traffic (HOA), oxygenated OA (OOA) that is typically associated with secondary OA (SOA), formed through the oxidation of organic vapor precursors or heterogeneous processes. The model is not capable of identifying the main SOA precursors, but often differentiates OOA based on its volatility and degree of oxygenation (semi-volatile fraction: SV-OOA and low-volatility fraction: LV-OOA) due to the available highly time-resolved data.

However, the cost and operational requirements of the AMS make its deployment impractical throughout a dense monitoring network and over longer time periods. As a result, most available datasets are often limited to few weeks of measurements and factors are extracted mainly based on diurnal variations in POA emission strength and SOA oxygen content (Zhang et al., 2011; El Haddad et al., 2013). Highly mobile measurements on platforms as aircrafts (e.g. DeCarlo et al., 2008) or vehicles (e.g. Mohr et al., 2011) are designed for regional studies, but are even more limited by cost, availability and time than stationary studies. This hinders the determination of the aerosol regional and seasonal characteristics and evaluation of long-term emission trends, limiting the information required for model validation and development of efficient mitigation strategies. Furthermore, the negligible transmission efficiency of the AMS inlet for coarse particles, prevents the characterization of their chemical nature and contributing sources.

The recent development of the aerosol chemical speciation monitor (ACSM, Ng et al., 2011, Fröhlich et al., 2013) has enabled the establishment of dense networks of long-term AMS-type measurements and source apportionment of the organic aerosol (e.g. Crippa et al., 2014 using AMS for shorter campaigns within the EUCAARI project or EMEP/ACTRIS projects for longer multi-season campaigns using ACSM). However, the mass spectrometers used by the ACSMs have far lower mass resolution than the AMS, reducing their performance for OA characterization and source apportionment. An alternate monitoring strategy involves extending AMS spatial and temporal coverage by measuring the nebulized water extracts of filter samples (Daellenbach et al., 2016, restricted to WSOA in Mihara et al., 2011). This approach allows the retroactive investigation of specific events, e.g. haze events in China (Huang et al., 2014) and AMS measurements of coarse mode



aerosol (Bozzetti et al., 2016a, b, in prep.). Additionally, such offline filters are routinely collected and are already available over multi-year periods at many air quality monitoring stations around the world for years/decades. Unlike single-season online AMS studies, the offline AMS analysis of filter samples may reveal seasonal and long-term variations in the emissions of POA and SOA precursors, required for model validations and the establishment of efficient mitigation

strategies.

Here, we present offline AMS measurements of PM10 (particulate matter with an aerodynamic diameter smaller than 10 µm) at 9 stations in central Europe with different exposure characteristics for the entire year of 2013 (819 samples). The sites cover rural and urban locations, including urban background and traffic and wood-burning influenced stations. Such long-

term multi-site analyses allow the quantitative description of the temporal and spatial variability of the main OA sources and may provide further insights into SOA precursors and formation pathways. This paper focuses on the identification of the main factors influencing the OA concentrations at the different sites and the assessment of the associated uncertainties without analysing the other aerosol components in detail. In the second part, we will investigate the site-to-site differences and time series.

## 15  2    Methods

### 2.1    Study area and aerosol sampling

PM10 samples were collected at 9 sites in Switzerland and Liechtenstein (Tab. 1 and Fig. 1). 7 of the sites (Basel, Bern, Payerne, Zurich, Frauenfeld, St. Gallen, Vaduz) are located in northern Switzerland and Liechtenstein and 2 (Magadino and San Vittore) in southern Switzerland. Aerosol was sampled at the selected sites every $4^{th}$ day for 24h throughout the year

2013 onto quartz fibre filters (14.7 cm) using HiVol samplers (500 l min$^{-1}$). Filters were then wrapped in aluminium foil or lint-free paper and stored at -20°C. Field blanks were collected following the same approach.

### 2.2    Offline AMS analysis

The offline AMS analysis summarized below was carried out following the methodology developed by Daellenbach et al. (2016). For each analyzed filter sample, 4 16 mm diameter filter punches were sonicated together in 10 mL ultrapure water

(18.2MΩcm, total organic carbon TOC/ < 5 ppb, 25 °C) for 20 min at 30 °C. Liquid extracts were then filtered (0.45 µm) and nebulized in synthetic air (80% Vol N$_2$, 20% Vol O$_2$, Carbagas, Gümligen CH-3073 Switzerland) using a customized Apex Q nebulizer (Elemental Scientific Inc., Omaha 24 NE 68131 USA) operating at 60°C. The resulting droplets were dried using a Nafion® dryer, and then injected and analyzed by the HR-ToF-AMS. Three types of measurements were performed: (i) filter samples, (ii) field blanks (collected and treated in the same way as the exposed filters), and (iii)

measurement blanks (nebulized ultrapure water without filter extract). The measurement blank was determined before and after every filter sample or field blank. Each sample was recorded for 480 seconds (AMS V-mode, $m/z$ 12-447), with a





collection time for each spectrum of 30 seconds. Ultrapure water was measured for 720 seconds. Once per day, ultrapure milliQ water was nebulized with a particle filter interposed between the nebulizer and the AMS, providing the gas-phase contribution to the measured mass spectrum, which was then subtracted during analysis from both blanks and filter samples. The filters from Zurich were analysed twice with a time difference of approximately 5 months to assess the measurement

repeatability. High resolution mass spectral analysis was performed for each *m/z* (mass to charge) in the range of 12- 115. The interference of $NH_4NO_3$ on the $CO_2^+$ signal described by Pieber et al. (2016) was corrected as follows (Eq. 1):

$$CO_{2,real} = CO_{2,meas} - \left(\frac{CO_{2,meas}}{NO_{3,meas}}\right)_{NH_4NO_3,pure} * NO_{3,meas} \tag{1}$$

The correction factor $\left(\frac{CO_{2,meas}}{NO_{3,meas}}\right)_{NH_4NO_3,pure}$ was determined based on measurements of aqueous $NH_4NO_3$ conducted

regularly during the entire measurement period (Pieber et al., 2016, from $\left(\frac{CO_{2,meas}}{NO_{3,meas}}\right)_{NH_4NO_3,pure} < 1\%$ and up to

10 approximately 5%).

## 2.3 Other chemical analysis

Organic and elemental carbon (OC, EC) content were measured by a thermo-optical transmission method with a Sunset OC/EC analyzer (Birch and Cary, 1996), following the EUSAAR-2 thermal-optical transmission protocol (Cavalli et al., 2010). Water-soluble carbon was measured by water-extraction followed by catalytic oxidation, non-dispersive infrared

detection of $CO_2$ using a total organic carbon analyser, only for the samples from Magadino and Zurich. Water-soluble ions ($K^+$, $Na^+$, $Mg^{2+}$, $Ca^{2+}$, $NH_4^+$; and $SO_4^{2-}$, $NO_3^-$, $Cl^-$) and methane sulfonic acid were analyzed using ion chromatography (Piazzalunga et al., 2013 and Jaffrezo et al., 1998). Levoglucosan measurements were performed by a high-performance anion exchange chromatography (HPAEC) with pulsed amperometric detection (PAD) using an ion chromatograph (Dionex ICS1000) following Piazzalunga et al. (2010 and 2013a). Free cellulose was determined by an enzymatic conversion to D-

glucose (Kunit and Puxbaum, 1996) and subsequent determination of glucose with HPAEC (Iinuma et al., 2009). Online measurements of gas-phase compounds and meteorology were also performed at selected sites.

## 3 Source apportionment

### 3.1 General principle

Source apportionment of the organic aerosol is performed using positive matrix factorization (PMF, Paatero, 1994). PMF is a

25 statistical un-mixing model explaining the variability of the organic mass spectral data ($x_{i,j}$), as linear combinations of static factor profiles ($f_{j,k}$) and their time-dependent contributions ($g_{i,k}$), se Eq. 2. The index *i* represents a specific point in time, *j* an ion, and *k* a factor. The elements of the model residual matrix are termed $e_{i,j}$.




$$x_{i,j} = g_{i,k} \times f_{k,j} + e_{i,j} \tag{2}$$

In the input data matrix, each filter sample was represented on average by 11 mass spectral repetitions to examine AMS measurement repeatability on the PMF outputs. A corresponding preceding measured blank from nebulized ultrapure water

was subtracted from each mass spectrum. The input errors $s_{i,j}$ required for the weighted least-squares minimization by the model include the blank variability ($\sigma_{i,j}$) and the uncertainty related to ion counting statistics and ion-to-ion signal variability at the detector ($\delta_{i,j}$ Allan et al., 2003; Ulbrich et al., 2009). We applied a minimum error according to Ulbrich et al. (2009), and a down-weighting factor of 3 to all fragments with an average signal to noise lower than 2 (Ulbrich et al., 2009). Input data and error matrices included 202 organic ions and were rescaled by the estimated organic matter (OM) concentration,

calculated as the product of the OC concentrations measured by the sunset and the OM/OC ratios from the AMS measurements.

The Source Finder toolkit (SoFi v.4.9, Canonaco et al., 2013) for Igor Pro software package (Wavemetrics, Inc., Portland, OR, USA) was used to configure the PMF model and for post- analysis. The PMF algorithm was solved using the multilinear

engine-2 (ME-2, Paatero, 1999), which enables an efficient exploration of the solution space by a priori constraining the $f_{k,j}$ elements ($n$ and $m$ being any two realizations of $j$) within a certain range defined by the scalar a ($0 \leq a \leq 1$), such that the modelled $f_{k,j}$' satisfy Eq. 3:

$$\frac{(1-a)*f_{k,n}}{(1+a)*f_{k,m}} \leq \frac{f_{k,n'}}{f_{k,m'}} \leq \frac{(1-a)*f_{k,n}}{(1+a)*f_{k,m}} \tag{3}$$

In all PMF runs (unless mentioned otherwise), we used the high resolution mass spectra for HOA and COA (cooking OA) from Crippa et al., (2013b) as constraints. Fitted ions in our datasets missing in the reference profiles were inferred from published unit mass resolution (UMR) profiles (Ng et al., 2011 and Crippa et al., 2013c). The relative intensity allocated to a missing ion was the intensity of the corresponding UMR peak minus the intensities of all HR ions present in the HR reference profile. For these ions, an $a$-value of unity was set.

Source apportionment analysis was performed following the scheme shown in Fig. 2 and discussed below. Exploratory unconstrained and constrained PMF runs provided information on the number of interpretable factors (Sec. 3.2). Multiple constrained PMF runs were then performed, to assess the model sensitivity to the chosen $a$-value, the model starting point and input matrix (entire dataset: PMF$_{block}$, only Zurich: PMF$_{zue,isol}$, 1 filter per site and month: PMF$_{1filt/M}$, repeated

measurements for Zurich: PMF$_{zue,reps}$) and repeated measurements (Sec. 3.3). The obtained factors were then classified and corrected for their recovery (Sec 3.4 and 3.5). Finally, the different solutions were evaluated and only the solutions that satisfied a set of predefined criteria (Sec. 3.6 and supplement) were considered.

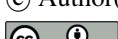



### 3.2    Preliminary PMF

We explored constrained PMF solutions, ranging from 1-10 factors. This investigation is performed on the entire dataset, including all stations and seasons (details in SI). The impact of the number of factors on the residuals is examined in the supplement. The introduction of 2 factors, in addition to HOA and COA, resulted in a significant reduction in the residuals and the separation of BBOA and OOA contribution. BBOA exhibited a prominent seasonal variation with a significant increase during winter and contributed most to the explained variation of the fragment $C_2H_4O_2^+$, originating from the decomposition of anhydrous sugars, i.e., from cellulose pyrolysis. OOA was identified based on its mass spectral fingerprint, with high contribution from oxygenated ions at $m/z$ 43 and 44. A further increase in the number of factors did not significantly contribute to the reduction in the residuals. However, the introduction of a 5$^{th}$ factor allowed the separation of the OOA into two different factors, with distinct seasonal variability and different relative contributions from oxygenated fragments at $m/z$ 43 and 44. The two OOA factors will be referred to as winter and summer OOA (WOOA and SOOA), according to their seasonality. The introduction of a 6$^{th}$ factor allowed resolving a factor with a distinct time series explaining the variability of sulfur-containing fragments (e.g. $CH_3SO_2^+$). This factor will be referred to as sulfur containing organic aerosol (SC-OA). We explored higher order solutions, but could not interpret the resulting factor separations. Therefore, we further consider a 6-factor solution below.

### 3.3    Sensitivity analysis

We assessed the model sensitivity to the chosen $a$-value for HOA and COA, the model starting point (independently for all four PMF inputs, as described below). The $a$-values were independently varied for HOA and COA ($a$-value from 0 to 1 with increments of 0.1, giving 121 $a$-value combinations). For every $a$-value combination, the model was initiated from five different pseudo-random starting points (seeds), yielding 605 total runs. As the selection of the $a$-value combination was randomized, the process was repeated four times in order to ensure that every $a$-value combination was represented at least once (2420 runs), which in turn provided an assessment of the seed effect on the results.

While this approach has been proven very effective in selecting a range of environmentally relevant solutions (Elser et al., 2016a, 2016b and Daellenbach et al., 2016), the resulting modelling errors may be underestimated. Paatero et al. (2014) compare the effectiveness in estimating modelling errors using two different approaches: the displacement (DISP) and bootstrap analysis (BS), respectively. BS involves applying the model to input matrices consisting of a subset of the entire dataset. DISP involves running PMF several times using randomly perturbed factor profile elements of a reference solution, but allowing a defined difference in Q from the reference solution. Both approaches are computationally intensive, especially DISP. Because of such computational limitations the combination of BS and DISP was not feasible for the dataset presented here, especially in combination with $a$-value sensitivity tests. Therefore, we chose to perform 4 sensitivity tests performing



PMF runs using 4 different input datasets, presented in the following. These sensitivity tests allow conclusions on the stability of PMF analysis when reducing the temporal or spatial resolution as well as the influence of the measurement repeatability.

1. $PMF_{block}$: PMF was performed on data from all seasons and all sites combined (all measured in October 2014). The corresponding data and error matrices involved 819 samples from 9 sites with 202 ions and on average 11 spectra per sample. This represents the base case.
2. $PMF_{zue,iso}$: PMF was performed on data from Zurich alone (isolated from $PMF_{block}$ input). The corresponding data and error matrices involved 91 samples with 202 ions and on average 11 spectra per sample.
10
3. $PMF_{1filt/month}$: PMF was performed on data from all sites but only considering the 1st filter collected for every month (12 filters/site), as for these samples levoglucosan and cellulose data was available. The corresponding data and error matrices involved 108 samples with 202 ions and on average 11 spectra per sample.
4. $PMF_{zue,reps}$: PMF was performed on data from the repeated measurements of Zurich samples. The corresponding data and error matrices comprised 91 samples with 196 ions and on average 14 spectra per sample.

For each of the four PMF datasets, 2420 PMF runs were performed for evaluating the sensitivity of the model to the chosen $a$-value and the seed.

### 3.4 Factor classification

From the sensitivity analysis, a large number of solutions were generated. Systematic analysis of these solutions required
20 automatic identification/classification of the retrieved factors within each solution. We applied a sequential classification algorithm as follows. The identity of HOA and COA were identified first as their mass spectra were initially constrained. In a second step, the factor showing the highest explained variation for $C_2H_4O_2^+$ among the 4 remaining factors was identified as BBOA. In a third step, the factor with the highest explained variation for $CH_3SO_2^+$ among the 3 remaining factors was identified as SC-OA. From the last two factors, the one with the highest explained variation of $CO_2^+$ was identified as
25 WOOA and the other as SOOA.

### 3.5 Recovery and blank corrections

After factor identification, factor time series are corrected using factor-specific recoveries (Eq. 4, resulting in $OA_{i,k}$) determined in Daellenbach et al. (2016) for HOA, COA, BBOA, and OOA.

$$OA_{i,k} = \frac{\frac{G_{i,k}}{R_k}}{\sum_k \frac{G_{i,k}}{R_k}} * OA_i \qquad (4)$$



where $g_{i,k}$ are the concentrations of factor $k$ at the timepoint $i$, $R_k$ the recoveries of the respective factor and $OA_i$ the OA concentration. The contributions of different factors to the field blank samples were estimated by inserting in the input matrix the field blank mass spectra. This was performed only for a limited number of runs, using the PMF$_{block}$ dataset. The factor contributions to the field blanks were used to correct the output factor time series, and the uncertainty induced by the blank subtraction was propagated.

### 3.6    Solution selection

PMF solutions were evaluated based on their factor profiles, time series, and the OC mass closure. Solutions were selected if they satisfied the following set of criteria:

1.  $f CO_2^+ < 0.04$ in HOA and COA factor profiles (HOA based on Aiken et al., 2009, Mohr et al., 2012, Crippa et al., 2013b, 2014 and COA based on Crippa et al., 2013b, 2013c, Mohr et al., 2012);

2.  $f C_2H_4O_2^+ < 0.004$ and 0.01 in HOA and COA, respectively (HOA based on Aiken et al., 2009, Mohr et al., 2012, Crippa et al., 2013b, 2014 and COA based on Crippa et al., 2013b, 2013c, Mohr et al., 2012);

3.  HOA correlates significantly with NOx being the sum of NO and NO$_2$ (defined below);

4.  HOA correlates significantly better with NOx than COA; BBOA correlates significantly with levoglucosan (defined below);

5.  SC-OA correlates significantly with $CH_3SO_2^+$ (defined below);

6.  for samples from Zurich and Magadino, where WSOC data are available, modelled and measured OC mass are comparable for a different set of conditions (see below and in SI).

The first two criteria (1-2) insure an appropriate separation of HOA and COA from OOA and BBOA, respectively. Criteria 3-6 relate to the evaluation of the correlation between factor and marker time series. This was achieved by computing the Fisher-transformed correlation coefficient $z$ at different stations (Eq. 5):

$$z = 0.5 * \ln\left(\frac{1+r}{1-r}\right) = \arctan(r) \qquad (5)$$

where $r$ is the correlation coefficient between factor and marker at a given station. The obtained $z$ values at the different stations are subsequently averaged before further analysis. A $t$-test is then used to verify the significance ($\alpha$=0.5) of the average correlation coefficient between factor and marker time series (Eq. 6):

$$t_{avg} = \frac{r_{avg}}{\sqrt{\frac{1-r_{avg}^2}{N-2}}} \qquad (6)$$





Here, $r_{avg}$ is the correlation coefficient averaged over the different stations, derived from the average $z$ value, $t_{avg}$ is the corresponding $t$-value and $N$ is the average number of samples at the different stations. To evaluate whether HOA correlated significantly better with NOx than COA did, the average $z$ values obtained between HOA and NOx and between COA and NOx were compared, using a standard error on the $z$ distribution of $1/\sqrt{N-3}$ (Zar, 1999).

The last criterion relates to OC mass closure. A Monte Carlo approach was applied to evaluate whether a combination of water soluble factor time series and recovery parameters would achieve OC mass closure, as described in the following. For samples from Zurich and Magadino, offline AMS measurements were scaled to the water soluble organic matter (WSOM), calculated using WSOC measurements and OM/OC ratios from the AMS HR analysis. The water-soluble contributions from an identified aerosol source in a sample $i$ were rescaled to its total organic matter concentration ($OA_{i,k}$), where k represents a given factor, using combinations of factor recoveries as determined by Daellenbach et al. (2016). For SC-OA, whose recovery was not previously determined, a recovery value was stochastically generated between 0 and 1. The obtained $OA_{i,k}$ concentrations were then converted to organic carbon concentrations $OC_{i,k}$, using factor specific OM/OC ratios determined from the factor profiles. The sum of $OC_{i,k}$ from all factors $k$ (mod-$OC_i$) was then evaluated against the measured OC (meas-$OC_i$). For this, the residual OC mass (res-$OC_i$) for each sample is calculated (meas-$OC_i$ – mod-$OC_i$), and the residual distributions were examined for different conditions that are thoroughly described in the Supplement.

For each of the Monte Carlo simulations, the water soluble factor time series satisfying criteria 1-6 were used together with a combination of factor recoveries from Daellenbach et al. (2016) as input data. The water soluble OC used for scaling $G_{i,k}$ matrix and the meas-$OC_i$ used for residual calculation were varied within their uncertainties (5%) and biases (5%) assuming a normal distribution of the errors. Likewise, constant biases were also introduced into the initial recovery distributions from Daellenbach et al. (2016). Monte Carlo simulations were performed and simulations for which res-$OC_i$ distributions were significantly different from 0 ($Q_{25}<0<Q_{75}$, details in SI) were discarded until 500 acceptable simulations were found. Factor time series and recovery parameters from all retained simulations were then averaged and the standard deviation of these averages represent our best estimate of the uncertainties for the single PMF datasets. The Monte Carlo process was repeated for the four different PMF datasets described above and the resulting average time series of their estimated uncertainties were compared.

## 4    Results and discussions

In this section, the final source apportionment results are presented and validated. The source signatures are presented in Fig. 3 for PMF$_{block}$ color-coded with the ion family. Fig. 4 shows the time series for Zurich obtained from all PMF approaches and Tab. 2 summarizes the correlation coefficients between factor and marker time series for Zurich (all PMF runs) and the





other sites in the study area (PMF$_{block}$) while the relation between factor and marker time series is displayed in Fig. 4 and 5. Presented are median (and quartile) results for all PMF runs accepted following the criteria described above.

### 4.1    Interpretation of PMF factors

***HOA:***    HOA profile elements were constrained using the reference profile from Crippa et al. (2013b). The final factor
profile (Fig. 3) maintains the same features, characterized by high contributions from hydrocarbon fragments. The fraction of oxygenated organic fragments that were missing in the initial reference profile, that were added based on UMR spectra, show an increased contribution to the ions above *m/z* 100 (see Sec. 3.1). While this indicates a possible overestimation of the contribution of these fragments, using this methodology, this increase does not substantially affect the results: e.g. HOA OM/OC ratio remains low (1.32, IQR 1.30-1.33). The HOA time series follows an expected pattern that matches the NO$_X$
yearly cycle (Fig. 4.a) except for San Vittore which is very likely due to the extremely high contribution of biomass burning at this site during winter, which may result in additional NO$_X$ inputs and/or may affect the separation of HOA by PMF. The HOA/NOx (Fig. 5.a) ratio at the different sites ($0.014 \pm 0.010$ µg m$^{-3}$ ppb$^{-1}$) lies within the range of literature values (0.001 to 0.028 (µg/m$^3$)/ppb, Lanz et al., 2007 and Kirchstetter et al., 1999). A similar average ratio was obtained for Zurich from the different sensitivity tests, but with high variability ($0.012\pm0.009$ µg m$^{-3}$ ppb$^{-1}$) similar to that obtained between the
different sites. This implies that the observed site-to-site differences are not statistically significant given our uncertainty in extracting HOA contributions.

***COA:***    COA profile elements were constrained using the COA profile Crippa et al. (2013b) and the obtained factor profile maintains the same features (OM/OC of 1.31, IQR 1.30-1.33, Fig. 3). For COA, no molecular marker is available for
validation purposes. Daellenbach et al. (2016) demonstrated that COA concentrations can be estimated with offline AMS (in Zurich at the same site) by constraining its signatures, but only with a high uncertainty. This was performed by comparing offline AMS results to those from a collocated ACSM, which owing to its higher time resolution enabled the identification of cooking emissions based on their diurnal cycles (Canonaco et al., 2013). Here, while no ACSM data were available, we followed the same methodology used in Daellenbach et al. (2016) to estimate the contribution of COA. The average COA
contributions estimated here and their yearly variability are similar to those from previous studies at the same sites, but as expected have high uncertainties (Fig. 4.b).

***BBOA:***    BBOA is identified based on its spectral fingerprint (OM/OC 1.74, IQR 1.74-1.75, Fig. 3), which, similar to previously extracted BBOA factors at other locations (Daellenbach et al., 2016, Lanz et al., 2007; Crippa et al., 2014),
exhibits high contributions from oxygenated fragments (CHO$^+$, C$_2$H$_4$O$_2^+$, C$_3$H$_5$O$_2^+$) from anhydrous sugars fragmentation (see comparison to nebulized levoglucosan in Supplement Fig. SII). Similar to levoglucosan, the BBOA time series shows an expected seasonal variation with high concentrations in winter, supporting the identification of this factor (Fig. 4.c). Except for Bern and Magadino (6.8 and 10.9), a similar ratio of BBOA to levoglucosan is found at all other sites (3.5 to 5), despite



apparent site-to-site differences in the model residuals during winter due to significantly higher contributions of BBOA at the southern stations (Fig. 5.b). The obtained ratios are within the range of values reported in literature (between 4 and 18 assuming OM/OC ratios between 1.6 and 1.8 for the non-AMS analyses, Zotter et al., 2014, Herich et al., 2014, Minguillon et al., 2011, Crippa et al., 2013a, and Favez et al., 2010). We note that a similar ratio is also found for the different PMF

datasets performed for the case of Zurich (BBOA/levoglucosan between 4.8 and 11.0). Taken together, the high (for most sites) correlation ($R^2$=0.78 for all sites, single sites in Tab. 2) between levoglucosan and BBOA and their consistent ratios at different sites and between the different PMF datasets indicates that BBOA is well resolved by PMF at all sites, despite potential site-to-site differences in BBOA composition.

*SC-OA:* Sulfur-containing fragments (e.g., $CH_3SO_2^+$) are predominantly apportioned to this factor, which also has a high OM/OC ratio (1.82, IQR 1.80-1.92, Fig. 3). As mentioned in Sec. 3.6, the recovery of SC-OA was unknown and had to be determined by mass closure, while the recoveries of the other factors were determined by comparison to their online counterparts (albeit for a different dataset, Daellenbach et al., 2016). In the lack of specific constraints (like an online counterpart), the recovery of SC-OA is highly uncertain and thus also the factor time series. A similar factor profile had been

extracted from previous online AMS datasets, and was related to the fragmentation of methane sulfonic acid (MSA) present in PM1 particles, a secondary product of marine origin (Crippa et al 2013b, Zorn et al., 2008). However, the SC-OA factor extracted here did not seem to be related to marine emissions because neither its variability nor its levels matched those of MSA (Fig. 4.d). First we compared the MSA levels measured in Zurich using ion chromatography to those estimated based on the concentration of sulfur containing fragments from offline AMS measurements in SC-OA (Eq. 7), based on Crippa et

al. (2013b):

$$MSA_{i,est} = SC - OA_i * \frac{f_{SC-OA}(CH_2SO_2^+) + f_{SC-OA}(CH_3SO_2^+) + f_{SC-OA}(CH_4SO_3^+)}{0.147} \qquad (7)$$

Here, $MSA_{i,est}$ is the estimated MSA concentration, $SC-OA_i$ the factor concentration of the sulfur-containing factor, $f_{SC-OA}(CH_2SO_2^+)$ and the following summands the fractional contributions of the respective organic fragment to SC-OA, while 0.147 is a scaling factor from Crippa et al. (2013b). The estimated MSA levels are 5 times higher than the measured MSA, indicating the presence of another source of sulfur-containing species. Second, unlike marine OA factors from previous online datasets (lower size cut-off, typically PM1), the SC-OA time series does not correlate with MSA ($R^2$= 0.03). While MSA concentrations show a clear enhancement during summer, the SC-OA time series exhibit a very weak seasonal

variability with slightly higher concentrations in winter. SC-OA exhibits instead low background levels episodically intercepted by remarkable ten-fold enhancements, especially at urban sites affected by traffic emissions (e.g. the SC-OA contribution is significantly higher at sites with higher yearly $NO_X$ average levels). The yearly average concentrations of SC-OA ($R_{s,SC-OA,NOx}$ =0.8, n=9, p=0.01) and also of different POA factors correlate with NOx ($R_{s,HOA,NOx}$=0.4, n=9, p=0.27),





COA ($R_{s,COA,NOx}$ =0.43, n=9, p=0.23). Besides that, the SC-OA time series also correlates with that of NOx (overall $R^2$=0.4, for sites in Tab. 2). While HOA and BBOA also correlate with NOx, both of the secondary factors, WOOA and SOOA, do not, supporting the hypothesis that SC-OA consists of locally emitted anthropogenic (primary) OA. The site-to-site differences in SC-OA concentrations and temporal behaviour suggest that this factor, which to the best of our knowledge is reported here for the first time, is influenced by primary sources.

***Oxygenated OA factors:*** Unlike oxygenated OA factors from limited-duration intensive online campaigns characterized by a high temporal resolution in which factor variability is thought to be primarily driven by volatility and/or local oxidation reactions, OOA factors are here resolved based on differences in their seasonal behavior: SOOA (in summer) and WOOA (in winter). The SOOA (summer) and WOOA (winter) mass spectral signature (Fig. 3) show similarities with OOA from earlier measurements (Ng et al., 2011, Canonaco et al., 2013, 2015), with high contributions of $C_2H_3O^+$, $CO_2^+$ and high OM/OC ratios though SOOA (OM/OC = 1.89, IQR 1.88-1.89) is less oxidized than WOOA (OM/OC = 2.12, IQR 2.11-2.15). The mass spectral fingerprints (Fig. 3), the temporal behaviour (Fig. 4e and f) and the relation to markers (Fig. 5c and d) of the two factors are in agreement with those from earlier work at other locations, including Zurich (Daellenbach et al., 2016), Payerne (Bozzetti et al., 2016a), and Lithuania (Bozzetti et al., 2016b). This OOA separation appears to be typical for PMF analysis of long-term, low time-resolution OA mass spectra of filter samples.

SOOA correlates significantly among the different sites (also south and north of the alpine crest) and with local temperature (Fig. 5c). The SOOA exponential increase with average daily temperatures from 5-30 °C is consistent with the exponential increase in terpenes emissions, which are dominant biogenic SOA precursors (Guenther et al., 2006). This is also consistent with the mass spectral fingerprint of this factor, characterized by an $fC_2H_3O^+$ of 0.10 and an $fCO_2^+$ of 0.13, which are similar to values reported for chamber SOA from terpenes or at an urban location (Zurich) during summer (Canonaco et al., 2015). A similar temperature dependence of biogenic SOA concentrations has been observed for a terpene-dominated Canadian forest (Leaitch et al., 2011) and for the case of Switzerland, using a similar source apportionment model (Daellenbach et al., 2016; Bozzetti et al., 2016a). Taken together, these observations suggest that SOOA principally derives from the oxidation of biogenic precursors during summer. Site-to-site SOOA concentrations were not statistically different within our model errors, assessed from the different sensitivity tests for the case of Zurich. Therefore, even though the behavior of SOOA at the different sites studied here might be controlled by various parameters, including tree cover, available OA mass, air mass photochemical age and oxidation conditions (e.g. $NO_X$ concentrations), temperature seems to be the main driver of the SOOA concentrations. Indeed, the aforementioned parameters may contribute, together with model and measurement uncertainties, to the observed scatter in the data. Biogenic VOC emissions might even in winter be non-negligible (Oderbolz et al., 2013; Schurgers et al., 2009; Holzke et al., 2006) and, therefore, significant winter-time SOOA concentrations are not in disagreement with the hypothesized biogenic origin. The lower SOOA concentrations in the temperature range between 7





and 12 °C might be explained by often occurring precipitation in this temperature range. We note that uncertainties related with this factor increase with decreasing concentrations (Fig. 7), due to the considerable contribution from other more significant wintertime sources (e.g. WOOA and BBOA). Furthermore, some winter-time SOA or other sources like primary biological OA (PBOA, see Sec. 4.2.2) might also mix into SOOA.

Compared to SOOA, the WOOA profile can be distinguished by a higher contribution from $CO_2^+$ and a lower $C_2H_3O^+$ (Fig. 3), similar to OOA factors previously extracted in this region during winter based on ACSM measurements. This fingerprint is characteristic of highly oxidized SOA from non-biogenic precursors with low H/C ratios (e.g. aromatic compounds from wood combustion emissions, Bruns et al., 2016). WOOA is well correlated with $NH_4^+$ (Fig. 5.d, overall $R^2$ 0.65 for all sites, overall $R^2$ 0.67 for all PMF runs for Zurich in Tab. 2) which is in agreement with earlier studies (e.g., Zurich in Lanz et al., 2008). This is probably explained by its correlation with other inorganic secondary ions $NO_3^-$ and $SO_4^{2-}$ (driven like WOOA by meteorological factors including boundary layer height and temperature), which govern the $NH_4^+$ concentration in the aerosol. WOOA exhibits a regional behaviour and its concentrations are correlated at all sites in the Swiss plateau. The WOOA mass spectral fingerprint, its seasonal variability and its high correlation with long-range transported anthropogenic inorganic secondary ions suggest that this factor is characteristic of a highly aged OOA influenced by wintertime anthropogenic emissions (e.g., biomass burning).

## 4.2 Uncertainty analysis

### 4.2.1 Model uncertainties

PMF uncertainties depend on the factor contribution. According to Ulbrich et al. (2009), reliable interpretation of factors with a low relative contribution is challenging. However, also the specificity of the time series and factor profile (caused by rotational ambiguity) and in this sense also solution acceptance criteria influence the uncertainty. In our analysis, we correct our results from WSOM to OM using $R_k$ and, thereby, introduce additional uncertainties (caused by the uncertainty of $R_k$ or an unknown $R_k$). The more uncertain $R_k$ is, the higher is the additional uncertainty in the extrapolation (Eq. 4). As mentioned in Sec. 3.5, $R_k$ constraints (recovery combinations for different factors) are available for $R_{HOA}$, $R_{COA}$, $R_{BBOA}$, and $R_{OOA}$ but not for $R_{SC-OA}$ and not for individual OOA factors (Daellenbach et al., 2016). With the available constraints of mass closure (for Magadino and Zurich), $R_{SC-OA}$ can only be determined with a high uncertainty (Fig. 6).

The variability of the factor time series for the single PMF sensitivity tests ($PMF_{block}$, $PMF_{zue,iso}$, $PMF_{1filt/month}$, $PMF_{zue,reps}$) is used as an uncertainty estimate (shaded area in Fig. 5). This estimate ($\sigma_a$) depends on the measurement repeatability (10 single mass spectra included for each sample) and on the selected PMF solution/ $R_k$ combinations and, therefore, also on the $a$-value. However, the variability depending (1) on the choice of input points (time and site; $PMF_{block}$, $PMF_{zue,iso}$, $PMF_{1filt/month}$) and (2) on the instrumental reproducibility ($PMF_{zue,reps}$) of the offline AMS measurements is not accounted for.





The contribution of (1) and (2) to the uncertainty is assessed through the sensitivity tests by examining the variability of the median factor time-series ($\sigma_b$). For the 12 filters common in all PMF datasets (PMF$_{block}$, PMF$_{zue,iso}$, PMF$_{1filt/month}$, PMF$_{zue,reps}$), we calculate a best estimate of the overall uncertainty (err$_{tot}$), by propagating both error terms: $\sigma_a$ and $\sigma_b$. As $\sigma_b$ is not available for all datapoints, we parametrized $\sigma_b$ as a function of the factor concentration (details in SI) and subsequently used this parameterized $\sigma_b$, $\sigma_b'$, to calculate an approximated overall error, err'$_{tot}$. err'$_{tot}$ is displayed in Fig. 7.b in comparison with $\sigma_a$(Fig 7.a). For all factors, err$_{tot}'$ are in general high (~a factor of 2), especially for low factor concentrations. It is worthwhile to note that for major factors exhibiting a similar seasonality, i.e. WOOA and BBOA, a great part of the uncertainty arises from $\sigma_b$ – the rotational ambiguity. By contrast, moderately soluble fractions, COA and HOA, constrained in the PMF the major part of err$_{tot}'$ is related to $\sigma_a$.

## 4.2.2   Influence of unresolved primary biological OA.

Unresolved sources in PMF are an inherent uncertainty of source apportionment analyses. As Bozzetti et al. (2016a) show, PBOA can present considerable contributions to OA in PM10 (constituting a large part of coarse OA). In the present analysis, this primary source could not be separated statistically using PMF (also not by using the mass spectral signature from Bozzetti et al., 2016) without mixing with other factors possibly because of the low water-solubility in combination with the absence of PM2.5 filters in the dataset. Since these coarse particles are only abundant in PM10 and not in PM2.5/PM1, the presence of both PM10 and PM2.5 samples, exhibiting a large gradient in PBOA, might allow an unambiguous separation of PBOA. In the following, we estimate the influence of PBOA in three alternative ways:

- Based on factor profiles: Bozzetti et al. (2016a) identified the AMS fragment $C_2H_5O_2^+$ as a possible tracer ion for PBOA. Based on the seasonality of SOOA (high in summer and low in winter), one can assume that SOOA in this study is a linear combination of PBOA and SOOA identified in PM2.5 and PM1. Based on the relative contribution of the ion $C_2H_5O_2^+$ to the factor profiles of SOOA from this analysis and literature profiles of PBOA and SOOA from Bozzetti et al. (2016a, study site: Payerne), we estimate that 17% of the water-soluble SOOA is in fact PBOA (between 5 and 22% for the different sensitivity tests). Using this approach, PBOA contributes during the warm months on average for the different sites 0.29 µg/m$^3$ with a site-to-site variability of 0.03 µg/m$^3$ (SOOA 1.71 µg/m$^3$ with a site-to-site variability of 0.16 µg/m$^3$, and OA 3.41 µg/m$^3$ with a site-to-site variability of 0.37 µg/m$^3$). This approach is very uncertain, mainly due to the uncertainty in PBOA and SOOA profiles, the assumption of a constant PBOA contribution to SOOA, and also the uncertainty of $R_{PBOA}$.

- Based on coarse OC: Bozzetti et al. (2016a) showed that coarse OC (OC$_{PM10}$-OC$_{PM2.5}$) is in summer dominated by PBOA for samples collected at a rural site in Switzerland (Payerne). For a subset of the samples used in the present work, OC in the PM2.5 fraction was also analysed (Basel, Bern, Magadino, Payerne, Zurich, accounting for 149 samples in total). For these samples, the OC$_{coarse}$ contribution to OC in the PM10 fraction is in summer 16% higher




than in winter (site-to-site variability of 4%). This part of OC might be related to resuspension caused by traffic or emissions of primary biological particles. However, the indicator ion $C_2H_5O_2^+$ shows higher concentrations with increasing $OC_{coarse}$ concentrations. Therefore, this increment can tentatively be ascribed to PBOA, which leads to a contribution of 0.55 µg/m³ to OC in summer (site-to-site variability 0.16 µg/m³). This results in an average summer

PBOA concentration of 1.21µg/m³ with a site-to-site variability of 0.39 µg/m³ when assuming an OM/OC of 2.2 (or 0.66±0.21 µg/m³, for OM/OC 1.2, OM/OC range according to Bozzetti et al., 2016). For Magadino (2014, Vlachou et al., in prep.), $OC_{coarse}$ represents 8% of OC in PM10 in winter while this ratio is 25% in summer. It can be assumed that the difference of 17% in summer can be attributed to PBOA. Extrapolating this estimate to the overall data set from 2013 considered in this study and assuming an OM/OC of 2.2, PBOA contributes in summer on

average 0.97 µg/m³ to OA in PM10, with a site-to-site variability of 0.13 µg/m³ (or 0.63±0.07µg/m³ OA with for OM/OC of 1.2).

- Based on cellulose: It has previously been shown that free cellulose contributes strongly to PBOA (25% of PBOA mass, for measurements made in Payerne during summer 2012/winter 2013; Bozzetti et al., 2016a). Therefore, we

can attempt to use cellulose analyses on a subset of samples (the same one as for levoglucosan but Bern, see Sec. 2.3) to estimate PBOA concentrations (Fig. 8). As seen for the case of $OC_{coarse}$, cellulose concentrations also increase with higher $C_2H_5O_2^+$ concentrations. For the sites with cellulose measurements available (all sites in the study but Bern), cellulose average concentrations of 0.17 µg/m³ (site-to-site variability 0.08 µg/m³, in the warm season 0.18±0.07 µg/m³) are observed which corresponds to 0.69 µg/m³ PBOA with a site-to-site variability of 0.34

20 µg/m³ (, in the warm season 0.77±0.29 µg/m³), using the ratio cellulose / PBOA from Bozzetti et al. (2016a). In this last study conducted during summer (15 days in June/July 2012), PBOA concentrations of 3 µg/m³ on average (with cellulose concentrations of 0.8 µg/m³) were estimated, which is clearly above the observation made here. However, Bozzetti et al. (2016a) assessed a shorter time period with diurnal resolution, instead of 1 sample per month as in the present work. Cellulose concentrations from other European sites during other years are consistent with the

25 results in this study (Sanchez-Ochoa et al., 2007; Yttri et al., 2011). In general, the background cellulose concentrations at the southern alpine sites are higher and also the temporal behaviour deviates from the one observed at the northern sites: the maximal concentrations are not reached in July/August but rather in May or October/November. The different seasonality might be caused by different agricultural procedures. The higher background concentrations of cellulose for the southern Alpine sites might be caused by interferences from wood

burning, which in the absence of glucose analyses cannot be excluded.

All these PBOA estimates (between 0.3 to 1.0 µg/m³ during the warm season) are consistently lower than reported in Bozzetti et al. (2016a), with a factor of 3 to 10 times lower depending on the site. One should keep in mind that these



estimates are based on limited data sets in both studies (30 samples in Bozzetti et al. (2016) while 12 samples from the same site in this study).

### 4.3 Factor relative contribution at different sites

In general, the seasonality of the factor time series is consistent for all the 9 sites in the entire study area (Fig. 9). In summer,
SOOA is the main contributor to OA, while in winter POA becomes more important though WOOA still contributes significantly. In comparison to the sites in northern Switzerland, OA in the southern alpine valleys is dominated by BBOA in winter, while in the north WOOA also plays a role. The different factors contribute 0.39±0.11 (HOA, average and site-to-site variability), 0.24±0.12 (COA), 1.25±1.70 (BBOA), 0.53±0.27 (SC-OA), 0.93±0.23 (WOOA), 1.12±0.12 (SOOA) µg/m$^3$ for all sites during the entire year (Tab. 3). In northern Switzerland, POA (HOA+COA+BBOA) contributes less to OA
(POA/OA=0.3) than in the southern alpine valleys where POA/OA is equal to 0.6. Among POA, BBOA is the most important, with 88% of POA in the south and 43% in the north. The higher relative contribution of BBOA to POA in the southern alpine valleys than at the northern sites supports the conclusion that the high BBOA concentrations (e.g. 2.26 µg/m$^3$ Magadino compared to 0.53 µg/m$^3$ Vaduz) are not only a consequence of the meteorological situation in the valleys (strong thermal inversion close to the valley ground) but mainly reflect the emission strength. SC-OA, which is possibly linked to a
local source of rather primary origin, shows clear site-to-site differences, with e.g. high concentrations at a traffic site in Bern (1.08 µg/m$^3$) and low concentrations at a rural site in Payerne (0.16 µg/m$^3$). SOOA, believed to have strong influences from biogenic SOA, shows consistently low concentrations at all sites for low temperatures (0.65±0.60 µg/m$^3$ at 5-15°C) and clearly increased concentrations under warmer conditions (4.60±1.35 µg/m$^3$ at 25-35°C).

### 5 Conclusion

Sources contributing to OA are quantitatively separated and their uncertainty estimated statistically at 9 sites in central Europe throughout the entire year 2013 (819 samples). Thereby, 3 primary (HOA, COA, BBOA) OA sources are separated from 2 secondary (WOOA, SOOA) categories and a yet unknown source explaining sulfur-containing fragments (SC-OA). BBOA exhibits clearly higher concentrations at the alpine valley sites in southern Switzerland than at the sites in northern Switzerland. SOOA, characterized by high concentrations in summer, shows a more than linear increase with rising
temperatures as is observed from biogenic VOC emissions and biogenic SOA concentrations. WOOA, dominant SOA category during winter, closely correlates $NH_4^+$. The influence of PBOA, not resolved by PMF, is estimated using among others cellulose analyses and could be an important contributor. Cellulose's temporal behaviour suggests maximal PBOA contributions in northern Switzerland during summer while at the southern alpine sites maximal concentrations are reached in spring/autumn.



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





**Table 1:Study sites with geographical location and classification**

| Site (station code) | Classification | General location | altitude |
|---|---|---|---|
| Basel, St. Johann (bas) | Urban/background | North of Alps/Swiss plateau | 308 m. |
| Bern, Bollwerk (ber) | Urban/traffic | North of Alps/Swiss plateau | 506 m. |
| Frauenfeld, (fra) | Suburban/background | North of Alps/Swiss plateau | 403 m. |
| Payerne (pay) | Rural/background | North of Alps/Swiss plateau | 539 m. |
| St. Gallen,(gal) | Urban/traffic | North of Alps/Swiss plateau | 457 m. |
| Zurich, Kaserne (zue) | Urban/background | North of Alps/Swiss plateau | 457 m. |
| Vaduz, Austrasse (vad) | Urban/traffic | North of Alps/alpine valley | 706 m. |
| Magadino, Cadenazzo (mag) | Rural/background | South of Alps | 254 m. |
| San Vittore, (vi) | Rural/traffic | South of Alps/alpine valley | 330 m. |



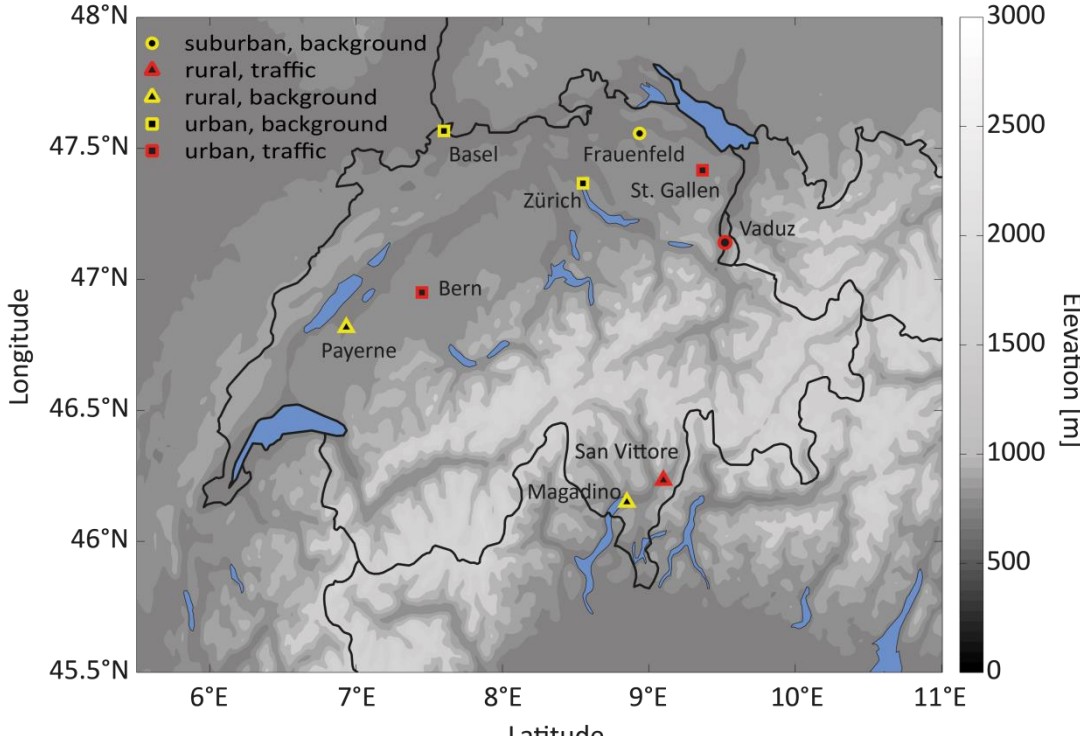

**Figure 1: Map of study area with locations of sites indicating their characteristics. The topography is displayed as meters above sea level.**





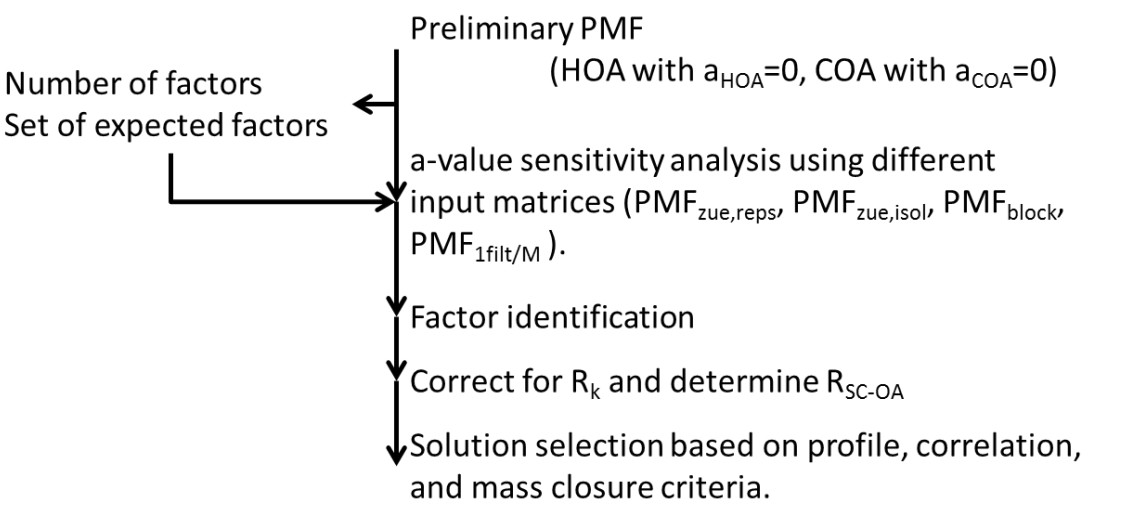

**Figure 2: Step-by-step outline of adopted source apportionment approach (factor recoveries $R_k$).**







**Figure 3: PMF factor profiles of HOA, COA, BBOA, SOOA, WOOA, SC-OA, color-coded with ion family of PMF$_{block}$ (average).**



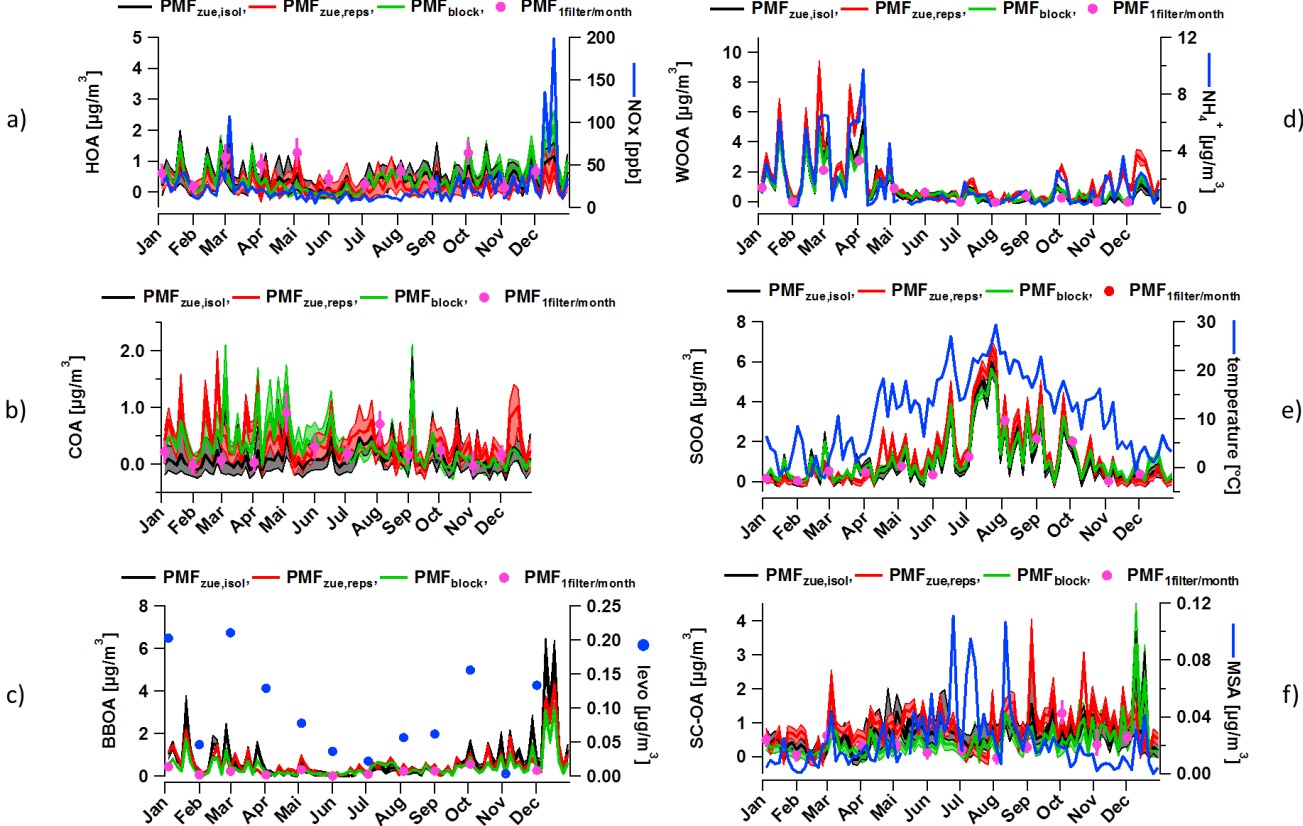

**Figure 4**: **HOA, COA, BBOA, SC-OA, SOOA, and WOOA and respective marker concentrations as a function of time for Zurich in 2013. Depicted are the results for the different PMF datasets (median) including the uncertainties (first and third quartile) (green: PMF$_{block}$, black: PMF$_{zue,isol}$, red: PMF$_{zue,reps}$, pink bullets: PMF$_{1filt/month}$).**



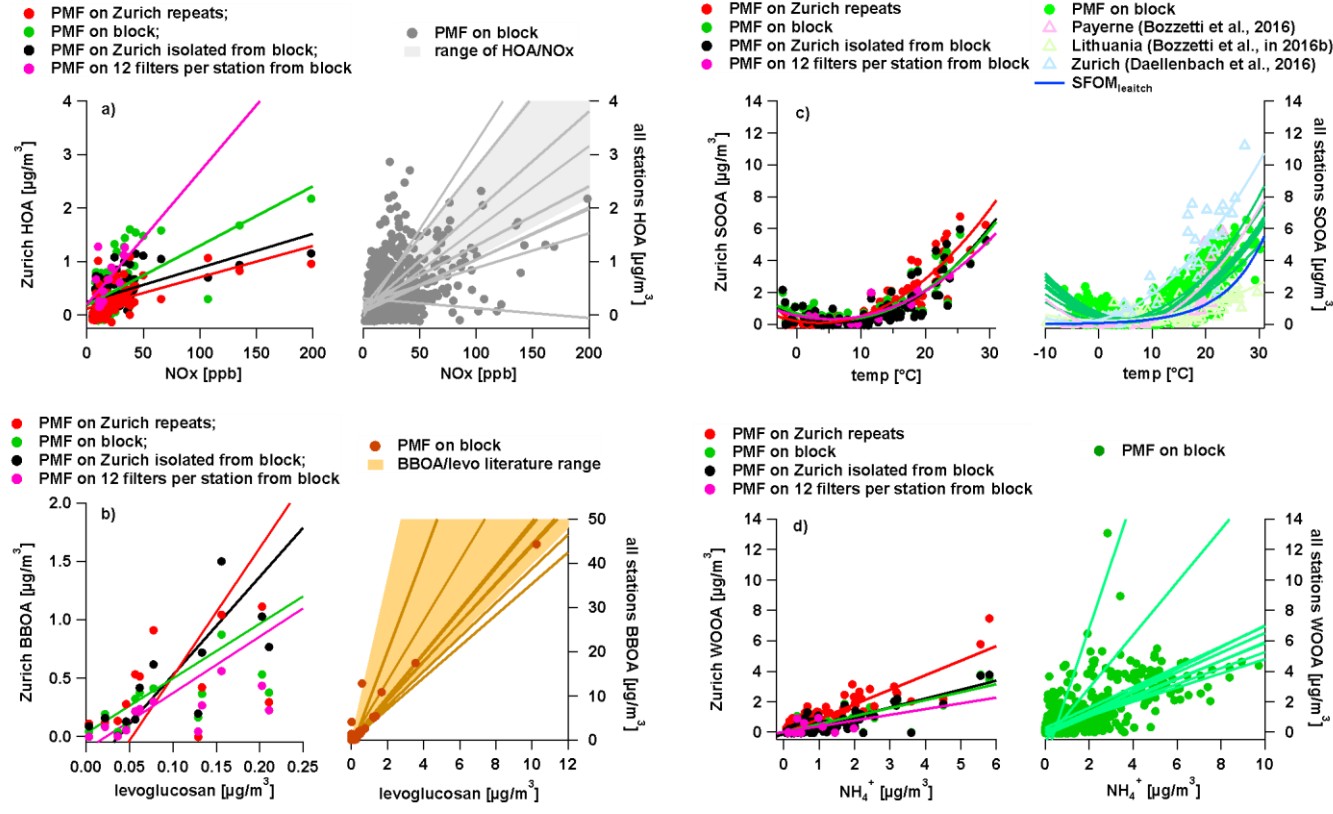

**Figure 5: Scatter-plots for the different extreme sensitivity tests for Zurich and for all sites for PMF$_{block}$ median concentrations): a) HOA vs NO$_x$, b) BBOA vs levoglucosan, c) SOOA vs temperature, d) WOOA vs NH$_4$.**





**Table 2: Comparison of factor time series to reference data for different PMF input datasets runs (by Pearson and Spearman correlation coefficient, $R_p$ and $R_s$). Displayed are the results for PMF$_{block}$ unless stated otherwise.**

| $R^2$ (number of points) | HOA vs NOx, $R_p^2$ | BBOA vs levo, $R_p^2$ | WOOA vs NH$_4^+$, $R_p^2$ | SOOA vs T, $R_s$ | SC-OA vs NOx, $R_p^2$ |
|---|---|---|---|---|---|
| Basel | 0.39 (91) | 0.91 (11) | 0.65 (91) | 0.65 (91) | 0.25 (91) |
| Bern | 0.23[1] (90) | 0.48 (12) | 0.53 (90) | 0.58 (90) | 0.20 (90) |
| Frauenfeld | 0.41 (89) | 0.74 (12) | 0.76 (90) | 0.60 (90) | 0.37 (89) |
| St. Gallen | 0.22 (91) | 0.40 (12) | 0.78 (91) | 0.68 (91) | 0.59 (91) |
| Magadino | 0.21 (91) | 0.54 (12) | 0.52 (91) | 0.66 (91) | 0.70 (91) |
| Payerne | 0.49 (91) | 0.67 (12) | 0.46 (91) | 0.63 (91) | 0.27 (91) |
| Vaduz | 0.39 (91) | 0.90 (12) | 0.77 (91) | 0.65 (91) | 0.54 (91) |
| San Vittore | 0.01 (90) | 0.99 (12) | 0.36 (90) | 0.72 (68) | 0.00 (90) |
| Zurich | | | | | |
|     PMF$_{block}$ | 0.38 (91) | 0.42 (12) | 0.79 (90) | 0.61 (91) | 0.50 (91) |
|     PMF$_{zue,iso}$ | 0.28 (91) | 0.59 (12) | 0.83 (90) | 0.63 (91) | 0.34 (91) |
|     PMF$_{zue,rep}$ (only 12 points) | 0.33 (12) | 0.23 (12) | 0.84 (12) | 0.85 (12) | 0.02 (12) |
|     PMF$_{1filt/m}$ | 0.30 (91) | 0.44 (12) | 0.78 (90) | 0.59 (91) | 0.63 (91) |

[1] 1 outlier removed.





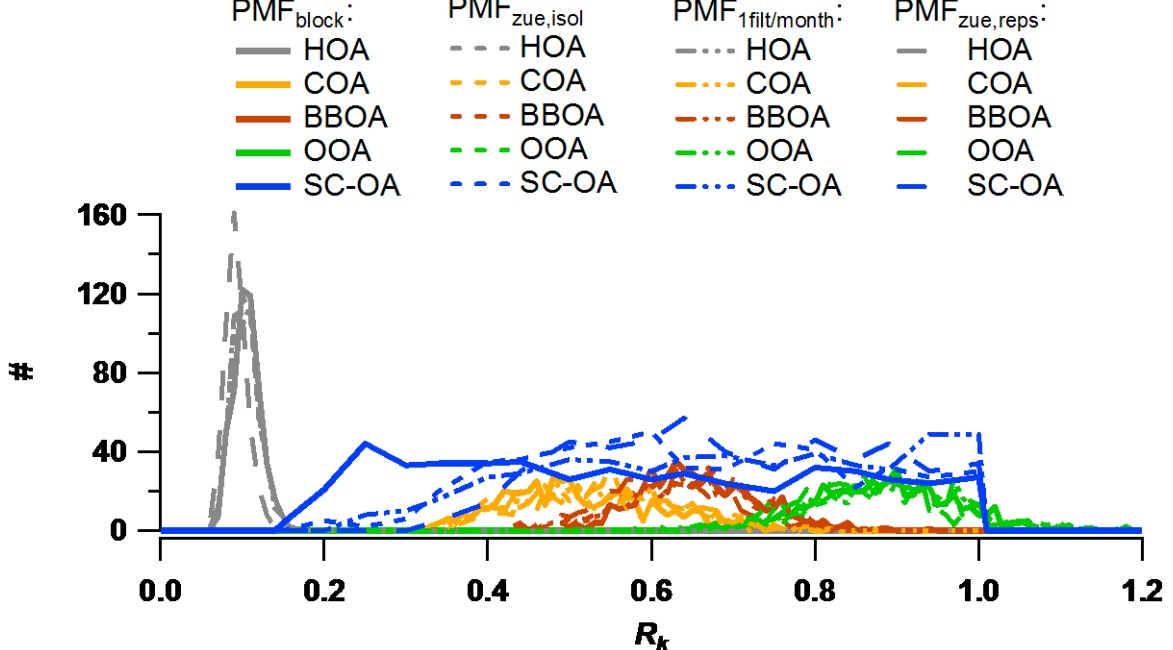

**Figure 6: Distributions of $R_k$ for HOA, COA, BBOA, OOA (WOOA plus SOOA) and SC-OA (500 pairs). A priori information for HOA, COA, BBOA, and OOA on $R_k$ are used from Daellenbach et al., 2016, with propagated errors and biases, while $R_{SC\text{-}OA}$ is determined in this study. Distributions of all factors have a resolution of d$R_k$=0.01 except for d$R_{SC\text{-}OA}$=0.05.**

**Figure 7**: Relative $\sigma_a$ (a) and err'$_{tot}$ (b) for factor concentrations > 0.1 μg/m³ as a function of factor concentration. err'$_{tot}$ includes the uncertainties from $a$-value/seed variability and $R_k$, and the different PMF datasets.





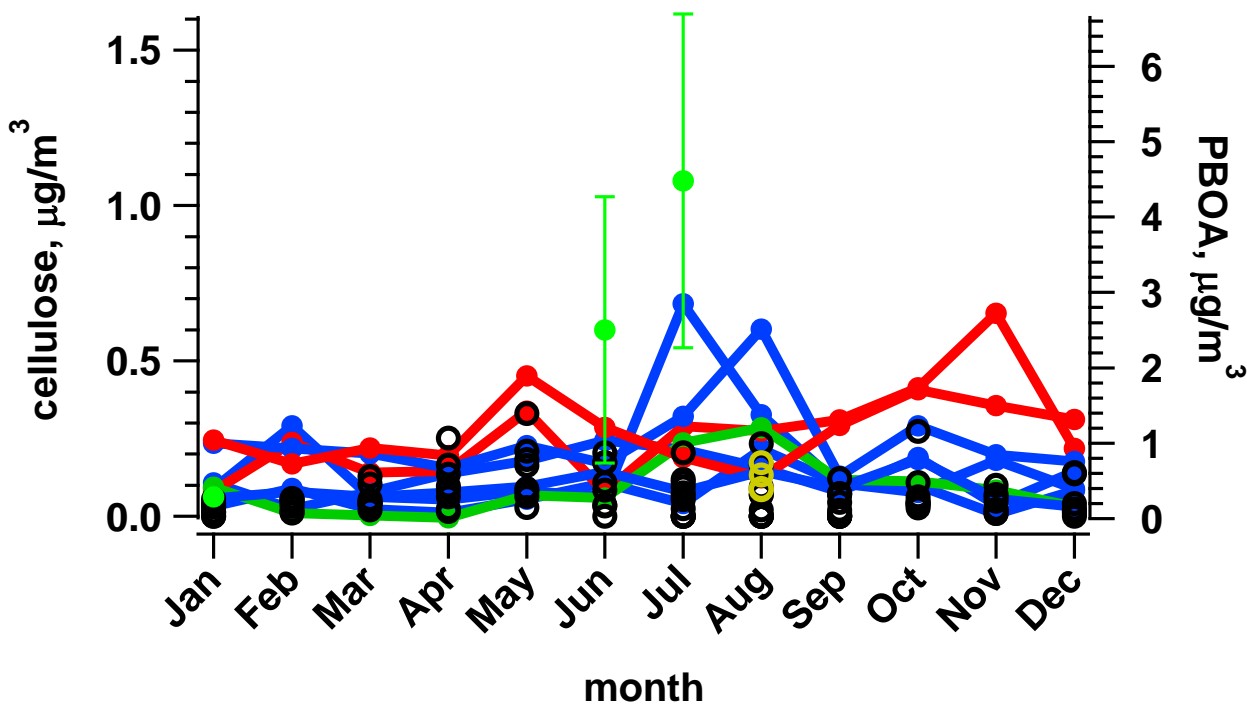

**Figure 8: Cellulose concentrations and PBOA concentrations (estimated based on cellulose concentrations) as a function of the season and site. For comparison literature data from other years is added European sites: Payerne (Bozzetti et al., 2016, error bars representingthe standard deviation of the measurements in June and July), Puy de Dôme, Schauinsland, Sonnblick, K-Puszta (Sanchez-Ochoa et al., 2007), Birkenes, Hyytiälä, Lille Valby, and Vavihill (Yttri et al., 2011).**





**Figure 9: Map of Switzerland with yearly cycles. Negative concentrations were set to 0 prior to normalization for display.**



**Table 3: Yearly average contribution and uncertainty of resolved factors for block PMF run for the different sites and the average for all sites. The uncertainty is calculated based on the variability of the yearly averages from PMF$_{block}$ and the variability between the 4 sensitivity tests.**

| Factor contribution and uncertainty µg/m$^3$ (%) | HOA | COA | BBOA | SC-OA | WOOA | SOOA |
|---|---|---|---|---|---|---|
| Basel | 0.55±0.21 (15±6) | 0.27±0.20 (7±5) | 0.61±0.20 (16±6) | 0.36±0.26 (10±7) | 0.89±0.37 (24±10) | 1.03±0.21 (28±6) |
| Bern | 0.52±0.22 (11±5) | 0.50±0.32 (11±7) | 0.55±0.19 (12±4) | 1.08±0.55 (23±12) | 1.01±0.43 (22±9) | 0.97±0.22 (21±5) |
| Frauenfeld | 0.47±0.20 (12±5) | 0.21±0.18 (5±5) | 0.55±0.19 (14±5) | 0.78±0.41 (20±10) | 0.81±0.34 (21±9) | 1.12±0.22 (28±6) |
| St. Gallen | 0.31±0.17 (10±6) | 0.1±0.15 (3±5) | 0.34±0.12 (11±4) | 0.53±0.29 (18±10) | 0.67±0.29 (23±10) | 1.03±0.21 (35±7) |
| Magadino | 0.35±0.17 (6±3) | 0.22±0.20 (4±3) | 2.26±0.74 (39±13) | 0.33±0.24 (5±4) | 1.35±0.55 (23±9) | 1.33±0.23 (32±4) |
| Payerne | 0.26±0.16 (9±5) | 0.10±0.16 (3±5) | 0.44±0.15 (15±5) | 0.16±0.17 (5±6) | 0.81±0.34 (28±12) | 1.18±0.21 (40±7) |
| Vaduz | 0.35±0.18 (10±5) | 0.19±0.18 (5±5) | 0.53±0.18 (15±5) | 0.67±0.36 (19±10) | 0.77±0.33 (22±9) | 1.03±0.21 (29±6) |
| S. Vittore | 0.28±0.16 (3±2) | 0.25±0.21 (3±2) | 5.50±1.78 (61±20) | 0.42±0.27 (5±3) | 1.26±0.52 (14±6) | 1.28±0.22 (14±2) |
| Zurich | 0.45±0.20 (12±5) | 0.31±0.22 (8±6) | 0.44±0.15 (12±4) | 0.48±0.30 (13±8) | 0.83±0.35 (23±10) | 1.16±0.22 (32±6) |
| Average | 0.39±0.18 (9±4) | 0.24±0.20 (5±4) | 1.25±0.41 (28±9) | 0.53±0.32 (12±7) | 0.93±0.39 (21±9) | 1.13±0.22 (25±5) |