# Peer review of "Long-term chemical analysis and organic aerosol source apportionment at 9 sites in Central Europe: Source identification and uncertainty assessment"

_Atmospheric Chemistry and Physics, 2017_

## Referee Comment (RC1) · Anonymous Referee #1 · 4 Apr 2017

The manuscript presents the results obtained from the off-line analysis of filter samples collected at 9 sites in central Europe with different exposure characteristics. The study is mainly focused on the identification of the different sources that contribute to the organic aerosol loadings using PMF analysis. The obtained results indicate that biomass burning is a major contributor to primary organic aerosol with 88% in the alpine valley and 43% north of the alpine crest. On the contrary, the sum of HOA, COA and BBOA contributes less at the sites north of the alpine crest than at the southern alpine valley. Secondary organic aerosol production is enhanced during summer due to the increased biogenic emissions with temperature. Finally, it is estimated that primary

biological particles which cannot be resolved by PMF could contribute significantly to PM10 organic aerosol.

The manuscript is very well written, coherent and easy to follow. A lot of effort has been put into the selection of the PMF solutions and the subsequent sensitivity analysis in order to provide the most sane and justified selection of factors, given the statistical nature of the analysis. This paper can be recommended for publication after some minor corrections listed below:

1) As the title reads "Long-term chemical analysis and organic aerosol source apportionment.." I would expect a short description of the trends in the chemical species as well of the 9 sites. It would be helpful as in a subsequent section the correlation with ammonium and nitrates is mentioned.

2) The possibility that WOOA could partially originate from the oxidation of BBOA could also be mentioned. During BB events ammonium is in a significant excess compared to sulfate, and this could possibly explain the good correlation between WOOA and $NH4+$.

Minor corrections:

Abstract L21: …smaller than 10 $\mu$m from 9 stations…

P16, L26: …closely correlates with $NH4+$.

---

## Referee Comment (RC2) · P. Paatero (Referee) · 6 Apr 2017

This work develops first a mathematical machinery whereby they attempt to formulate a small number of factor spectra (2 assumed "known" mass spectra plus 4 fitted mass spectra) whereby a large number (819) of mass spectra, measured in 9 locations in different times of the year, could be sucessfully fitted.

In the second part, they perform a very large number of repetitions of the modeling, so that different details of the model are varied. From variation of the results, they deduce reliability estimates for the obtained spectral profiles and contribution time series.

The mathematical operations are not adequately explained. It is impossible to know what they have done (and why) in different stages of the work. One of their key concepts is a constraining of F factor elements. Unfortunately, the defining equation (3) is so unclear (and possibly contains a typo) that I cannot even guess what they might mean by this equation.

This review is limited to the first part of the work, modeling the measured mass spectra by a bilinear model. Although the second part is important, and has required a large amount of work, it cannot be analyzed in the time that would be available for such analysis. Thus I do not comment on the second part.

I recommend that this manuscript should be published after the listed problems in mathematical presentation and elsewhere have been corrected.

Problems in the mathematical presentation

The fundamental principle of science is repeatability. Used methods should be defined clearly and in sufficient detail so that colleagues will be able (at least in principle) to repeat what was done by the authors. In the following, I discuss mathematics that have not been described in an understandable manner. In general, equations are the language of mathematics. Mathematical work should be described using equations. Verbal explanations may only help in understanding the equations, they cannot replace equations. In order to use equations, it is necessary to define symbols for various quantities. If necessary, use two-letter symbols. Consider typographic questions: The prime should be avoided. Complicated notation in subscripts or in superscripts is often very difficult to read.

P5 L9-11 (page 5, lines 9-11)

> .. by the estimated organic matter (OM) concentration,

> calculated as the product of the OC concentrations .. and

> the OM/OC ratios from the AMS measurements.

Reading this verbal explanation, it is very hard to understand what was done. In fact, I would need to try to write the equations in order to understand. Please do this equation writing for the benefit of your readers!

P5 EQ(3): I do not understand eq(3) at all.

- what are f_kn and f'_kn (or is it f_kn'), what difference is there between them?

- is there a typo in the equation? As shown, the equation appears impossible.

This is a key detail in the manuscript as it describes constraints that perhaps have never been published in similar work. It must be described so that it is understandable.

What was included in matrix X, to be modeled by PMF.

It remains unclear what information was contained in X. Section 2 lists a number of variables whose concentrations were measured. Were all these included in X, or were some of these included, or none of them?

P5 L8-9 say: Input data and error matrices included 202 organic ions..

Do you mean that input data consisted of 202 organic ions? The formulation you used may also be interpreted so that among other data, input data also included 202 organic ions.

Also regarding matrix X:

Input data ... were rescaled by the estimated organic matter (OM) concentration ...

In PMF, one is allowed to rescale data rows in any way, provided that error rows are also scaled in the same manner. Thus your scaling is OK. However, it would help the reader a lot if you state briefly why to scale, what advantage was achieved by scaling. Your scaling (which does not change profile spectra in any way) is useful for plotting figure 9. On the other hand, your scaling influences (improves or worsens) correlations between factor values and marker concentrations. Please explain and/or correct. Note

that this scaling does not change the computations performed according to Eq(4) in any way.

P5 L5-7 (input errors):

> The input errors ... include ... the uncertainty related to ion

> counting statistics and ion-to-ion signal variability at the detector.

I understand that you counted the ions. Then, ion-to-ion signal variability is *not* a source of uncertainty. If ion current is measured, instead of counting ions, then ion signal variablity *is* a source of error. Please correct or clarify.

P5 L20-24

describe a complicated method for filling knowledge gaps in the known/assumed fixed-factor spectra ("reference profiles") of HOA and COA. I am not fully convinced about the performance of this method. The "natural" alternative method is to leave the unknown elements in reference profiles (profiles of HOA and COA) as ordinary factor elements, to be fitted by ME-2 together with all "normal" non-fixed F factor elements, as explained in detail below. Please include this remark in the corrected ms, so that future colleagues are encouraged to follow the safer and simpler method instead of your complicated method.

Using "constrained factors" based on known profiles of HOA and COA.

This topic was very difficult to understand at first. It was not clear what is "constrained" by what. Now I assume that you mean the following: In all PMF runs, two constant F factors were used, i.e. two rows of factor F were defined as a-priori fixed, so that the values of these constant factors were set equal to previously known mass spectra of HOA and COA. Is this what you mean? – Using constant or constrained factors is not familiar to PMF users, not at all. Such unusual methodology must be carefully explained so that all readers have the possibility to understand what you have done. In particular, you should explain that using fixed factors is not the same as using inequalities in order to constrain factors to lie between upper and lower limits, set very close to each other. Also, you should go into technical details here, because it is possible to implement constant factors in two different ways in ME-2. You should guide your readers to the optimal usage. It is possible to use "constant factors" that reside in a different matrix, which is clumsy. The alternative is to keep all F factors in the same matrix but define that the elements in two first (say) rows of F are "locked", not allowed to change during the fitting process. These elements are set equal to the known profile values before initiating the fit. If there are gaps in the knowledge of HOA and COA, then those unknown elements in "locked" F rows should simply not be locked at all, so that they may obtain their best possible values during the fit. Use of constant factors or constrained factors often causes so-called "normalization conflicts". How did you protect your bilinear model against normalization conflicts? This is another important detail that should be communicated to colleagues who might follow your example.

P7 L16-17

> For each of the four PMF datasets, 2420 PMF runs were performed for

> evaluating the sensitivity of the model to the chosen a-value and the seed.

This statement mentions sensitivity of the model to random seed. The random seed determines the pseudorandom initial values of PMF fit. In plain language, this statement says that there were local solutions so that depending on seed, PMF iteration converged to different local solutions. Presumably, these solutions had comparable Q values because otherwise, Q values would be used for selecting between solutions. Now these different solutions are somehow pooled together and their presence is otherwise ignored.

The presence of multiple solutions should be properly reported (e.g. how many of PMF runs had multiple solutions, how many different individual solutions per PMF run were obtained at most and on the average, are the solutions rotationally equivalent having identical residuals of fit, etc.) There are no fixed rules on what to do with multiple solutions. On one extreme, it has been suggested that scientists may at will pick the one solution they like most and ignore the others. At the other extreme, PMF modeling of such data may be considered failing if there are several local solutions with comparable Q values.

If DISP is used for uncertainty estimation of a case with several local solutions, one often obtains the outcome that the model is "Not Well Defined" or "NWD". I would not suggest what the authors should do with their many-solution cases in addition to discussing them. Whatever they opt to do, they should describe it: what was done and why.

Section 3.3, sensitivity analysis

I cannot comment more on this analysis because I do not understand what the a-values are and how they were used.

P6 L26-28 say:

> Paatero et al. (2014) compare the effectiveness in estimating modelling errors

> using two different approaches: the displacement (DISP) and bootstrap analysis

> (BS), respectively.

Here seems to be a terminological problem: in the quoted paper, Paatero et al. estimated the uncertainties of estimated F factor elements in the situation when no modelling errors are present. These F uncertainties depend mainly on random error in X and on rotational freedom of factor matrices G and F. It was specifically emphasized that the obtained uncertainties do not cover effects of modelling errors in the results. (Examples of modelling errors: non-constant factor profiles, wrong uncertainties assigned to data values.) Thus modelling errors were not estimated in the quoted paper.

P6 L29:

> DISP involves running PMF several times using randomly perturbed

> factor profile elements of a reference solution

In fact, DISP estimation does not involve any randomness at all. F factor elements are pertubed in a systematic fashion by DISP. Perhaps here is confusion with Monte Carlo methods where random perturbations may be applied. DISP is not Monte Carlo.

In table (3), uncertainty estimates of percentage concentrations are not correctly computed.

Notation:

In supplement, subscript "i" is used as a subscript of Q. It is not defined what "i" means here. Does it mean number of factors? If yes, then the symbol used for number of factors should be used. If i does not have a definable meaning, then it might be clearer to omit the subscript in this case. In general, systematic use of subscripts would be a help for the reader. E.g. use i only as the index of sample (time), j as index of column of X, k index of factor. For other quantities, select other symbols and define what they mean.

In different places, factor elements are denoted as $G_{ik}$ and as $g_{ik}$.

This may confuse readers. Eventually they will recognize that this difference does not mean anything, but first they will waste time trying to understand. Either, select one notation (preferred), or, in the section "Notation" (to be written), specify that $G_{ik}$ and $g_{ik}$ mean the same, and also $F_{kj}$ and $f_{kj}$ mean the same.

Eq (2) is incorrect.

If you wish to use matrix element notation, then summation over k must be indicated. If you wish to use vector-matrix notation, then its use should be defined, especially because many of your readers are not familiar with such notation. In vector-matrix notation, index k would not be visible.

P5 L10: sunset –> Sunset OC/EC analyzer

P5 L21

> Fitted ions in our datasets missing in the reference ...

What do you mean by "fitted ions"? I do not understand this sentence.

P7 L21:

> The identity of HOA and COA were identified first as their mass spectra were

> initially constrained.

Why do you need to identify HOA and COA? I would assume that they are on pre-selected rows of F, such as rows 1 and 2. No identification is needed for factors that are in known positions. What is wrong here? Am I understanding all this completely wrong?

Supplement

Figure SI.1

shows ratios of obtained Q vs. expected Qexp. How were Qexp computed? Did you take into account that downweighted columns of X contribute very little to Qexp? How many downweighted columns were present?

Obtained ratios Q/Qexp are of the order of 10 for 6 factors. This indicates that there are one or several significant modeling errors. It has been assumed among atmospheric scientists that a ratio of 4, say, would not be significant, that it could be caused by random variations from the expected Q. This assumption is totally wrong. Such ratios (>1.5, say) always have a cause that preferably should be understood in a project where careful mathematical analysis is attempted.

Possible causes of modeling errors are: underestimation of random errors in mass spectra, variation of factor profiles with time and between sites, systematic errors in preprocessing m/z spectra, and spurious sporadic local sources that cannot be modeled by PMF. An attempt should be made at understanding and discussing those errors, even if the effect to obtained results cannot be eliminated any more at this final stage. One useful diagnostic is to examine contributions to Q from different m/z values, from different times of day and days of week, and so on.

Table SI.1

Why is there a table for mass closure criteria (used for rejecting bad solutions) when all entries in this table are identical? The criteria seem to concern distribution of residuals of OC fitting. Is it really so that the fit is rejected if 1st quartile point is negative and 3rd quartile point is positive? In other words, the fit is rejected if residuals are symmetrical around zero. Usually, such residuals would be considered desirable.

———————————————

---

## Referee Comment (RC3) · Anonymous Referee #3 · 10 Apr 2017

Comments on "Long-term chemical analysis and organic aerosol source apportionment at 9 sites in Central Europe: Source identification and uncertainty assessment" by Daellenbach et al. The manuscript presents new research which clearly fits within the scope of the journal. The text is well-written and fairly easy to follow. Some of figures, however, compile several information and are not as straightforward to interpret (e.g. Figure 8) – please make sure to modify them (color axis, split into subplots, etc.) to improve readability.

The technique described here is a follow-up of the characterization of OA measure-

ments based on filter collections followed by water extraction and analysis by HR-AMS, previously published, being the novelty a large statistics from 9 sampling sites and, most importantly, PMF analysis of the OA spectrum from filters. Although the former is unquestionably of scientific interest, it is the latter that will allow others to apply the technique and indeed reach its goals as described in the introduction. At its current stage, the manuscript doesn't fully achieve it.

Major comments:

* The description depth of the PMF applied to this very specific dataset doesn't seem to be proportional to its level of development in regard to the widely used techniques. Please detail it more carefully.

* Section 4.2.2 seems quite weak, three methods to estimate PBOA are presented, but no clear conclusion is given other than it underestimates based on previous literature. From my perspective this section doesn't add too much to the manuscript and could easily be removed, however if the authors wish to keep it, please make sure to better constrain the methods into a valid scientific output.

Minor comments:

* Abstract. L.21: add the word "from" between $\mu$m and 9.

* Abstract. L.24: remove "which is" and add a comma before related.

* P.2, L.31: Please remove "restricted to WSOA in".

* P.12, L.10: Remove the word "here".

* External gas-phase tracers (besides the use of NOx just to separate HOA, COA) could also add some information of the surrounding chemistry – for example, what is the ozone (over 24h, or just afternoon) in regard to SOOA and WOOA? And Ox?

---

## Referee Comment (RC4) · Anonymous Referee #4 · 18 Apr 2017

**General comments:**

In this paper the concentrations of the six types of organic aerosol (OA) components (HOA, COA, BBOA, WOOA, SOOA, and SC-OA) over Switzerland are reported based on the off-line analysis of the water-soluble aerosol components in aerosol samples using an aerosol mass spectrometer (AMS). The characteristics of the retrieved OA components, e.g., the relative abundances and seasonality, are presented. Further, the uncertainty of the concentrations of the retrieved OA is discussed. The source identification of OA components based on long-term samplings at multiple locations

is important, and the application of the aerosol mass spectrometry for the chemical analysis of aerosol samples collected on filters made it possible in this study. The contributions of the major sources of OA to the atmospheric concentrations in the studied area have been characterized well in view of location and seasonality.

Although the results presented in this paper are highly valuable, this paper needs substantial improvement in terms of the presentation quality. The explanations for the statistical analyses are not fully comprehensive, and a part of them would be flawed. Further, the point of this study is not very clear because both the methodology of the analysis itself and the results based on it are presented and discussed. To make the point clearer, it may be better to move the discussion on the uncertainty based on the results in Figures 6 and 7 to the experimental section or the supplement. Other minor issues regarding the presentation quality include inadequate explanations, undefined abbreviations/symbols, and grammatical errors.

For the reasons above, substantial improvement is required for the publication of this paper in its final form. More specific comments are listed below.

**Specific comments:**

Page 3, 1st paragraph: It may be better to explain more about previous source apportionment studies for organic aerosols using off-line AMS measurement techniques. The group of the first and corresponding authors reported two more studies, both of which were also for European sites (Bozzetti et al., 2017a, 2017b). There are also other source apportionment studies based on statistical analysis for the mass spectra obtained using off-line AMS techniques (Sun et al., 2011; Chen et al., 2016). Emphasis should be on which characteristics of atmospheric aerosols have not been studied tentatively even by the use the off-line AMS techniques.

Page 3, 2nd paragraph: The chemical analysis using the AMS was limited to the water-soluble component of organics in $PM_{10}$, although the water-insoluble organic component was also taken into consideration in the source apportionment. This point should

be addressed more explicitly.

Page 3, line 13: The site-to-site differences and time series are not explained in a specific part of this paper.

Page 4, lines 1-3: How were the mass spectra of the extracts from aerosol samples corrected for field blanks? Because the sensitivity of an AMS to aerosol components depends on the particle size, the signal intensity of organics should not be proportional to the organic mass flux from the nebulizer. For this reason, the assessment of the blank level is not straightforward. More explanation to this point is necessary.

Page 4, lines 9-10: The expression in the parenthesis is unclear and needs to be reworded.

Page 5, lines 9-11: The method for rescaling here and that explained in the 2nd paragraph of page 9 does not seem identical.

Page 5, equation 3: The constraint represented by equation 3 seems erroneous because the left and the right parts of the equation are identical.

Page 5, lines 21-22: Were the inferred fitted ions also for constraint? Does this sentence mean all the factors other than HOA and COA were inferred from published UMR profiles?

Page 5, lines 22-24: The explanation in this sentence is not clear. This sentence should be reworded.

Page 8, line 1: The values of the recoveries used in this study should be presented.

Page 8, line 2: The meaning of "the contributions of different factors to the field blank samples" is not clear. What was done is not clear, either.

Page 8, line 27: Is "$\alpha$=0.5" the significance level? Fifty percent is too high.

Page 8, line 26-28: How the statistical analysis using the average values from different

stations can be justified? The validity of this method is not obvious.

Page 9, lines 2-4: More details in the calculation should be given so that the readers can assess its validity.

Page 9, lines 14-16: Is the issue really explained in the supplement?

Page 10, line 2: What are the percentages of the accepted data?

Page 11, line33 – page 12, line 1: This sentence is not clear. Does COA relate to the discussion here?

Page 13, line 1: The "uncertainties" here should be relative uncertainties. This should be addressed explicitly.

Page 13, lines 2-3: The meaning of "contribution from other more significant wintertime sources" is not clear. Further, justification of the explanation in this sentence should be provided.

Page 13, lines 3-4: It is not clear why the mixing of some winter-time SOA into SOOA results in a larger uncertainty.

Page 14, lines 1-2: How was $\sigma_b$ calculated?

Page 14, line 8: The meaning of "$\sigma_b$ – rotational ambiguity" is not clear.

Page 14, lines 12-15: This sentence is not very organized and needs to be reworded.

Page 14, lines 24-26: What is the definition of the site-to-site variability? Was standard deviation calculated for the average values at respective sites?

Page 15, line 2: The use of the word "however" does not seem appropriate.

Page 16, line 5: Is POA here the sum of HOA, COA and BBOA? Shouldn't it be defined here instead of line 9?

Figure 2: The $a_{HOA}$ and $a_{COA}$ are not defined explicitly.

Figure 3: The definition of $f_{m/z}$ should be given.

Figure 9: The definition of $OA_{expl}$ is not given explicitly.

Page 2 (supplement): The relationship among "$Q_i/Q_{i,exp}$", "$\Delta(Q_i/Q_{i,exp})$", "$\Delta Q_i/Q_{i,exp}$", "($Q_i/Q_{i,exp}$ contribution)", and "$\Delta(Q_i/Q_{i,exp}$ contribution)" is not clear.

Page 3 (supplement): The definitions of "r(...)", "$Q_{25}(OC_{res})$" and "$Q_{75}(OC_{res})$" are not given.

Page 4 (supplement): The definition of "$f_{ion}$" is not given.

**Technical corrections:**

Page 1, line 21: Should "at" be added between "10 $\mu$m" and "9 stations"?

Page 3, line 20: "HiVol" should be spelled out.

Page 5, lines 29-30: The subscripts of "PMF" are not written consistently in the paper.

Page 10, line 18: Should "from" be added after "profile"?

Page 13, line 29: Should "Fig. 5" be "Fig. 4"?

Page 14, lines 30 and 33: Should "is in summer" be "in summer is"?

Page 14, line 33: "$OC_{coarse}$" should be defined in line 30..

Page 19, lines 9-10: The list of authors are incorrect.

Table 1: The commas after "St. Gallen" and "San Vittore" in the column "Site (station code)", and the periods after "m" in the column "altitude" should be omitted. The initial letter of "altitude" should be capitalized.

Figure 4 caption: It is better to write "HOA, COA,...." in the order of the corresponding panels.

Figure 5 caption: Should "$NH_4$" be "$NH_4^+$"?

Figure 7: Should "[" after "concentration" be "]"?

Page 4 (supplement): The "interquartile range PMF block" should be represented by a symbol because it is in a mathematical formula. It may be better to write "median bootstrap solutions" as the subscript of $\sigma$.

**References:**

Bozzetti, C., Sosedova, Y., Xiao, M., Daellenbach, K. R., Ulevicius, V., Dudoitis, V., Mordas, G., Byčenkienė, S., Plauškaitė, K., Vlachou, A., Golly, B., Chazeau, B., Besombes, J.-L., Baltensperger, U., Jaffrezo, J.-L., Slowik, J. G., El Haddad, I., and Prévôt, A. S. H.: Argon offline-AMS source apportionment of organic aerosol over yearly cycles for an urban, rural, and marine site in northern Europe, Atmos. Chem. Phys., 17, 117–141, doi:10.5194/acp-17-117-2017, 2017a.

Bozzetti, C., El Haddad, I., Salameh, D., Daellenbach, K. R., Fermo, P., Gonzalez, R., Minguillón, M. C., Iinuma, Y., Poulain, L., Müller, E., Slowik, J. G., Jaffrezo, J.-L., Baltensperger, U., Marchand, N., and Prévôt, A. S. H.: Organic aerosol source apportionment by offline-AMS over a full year in Marseille, Atmos. Chem. Phys. Discuss., doi:10.5194/acp-2017-54, in review, 2017b.

Chen, Q, Miyazaki, Y., Kawamura, K., Matsumoto, K., Coburn, S. C., Volkamer, R., Iwamoto, Y., Kagami, S., Deng, Y., Ogawa, S., Ramasamy, S., Kato, S., Ida, A., Kajii, Y., and Mochida, M.: Characterization of chromophoric water-soluble organic matter in urban, forest, and marine aerosols by HR-ToF-AMS analysis and excitation emission matrix spectroscopy, Environ. Sci. Technol., 50, 10,351–10,360, 2016.

Sun, Y., Zhang, Q., Zheng, M., Ding, X., Edgerton, E. S., and Wang, X.: Characterization and source apportionment of water-soluble organic matter in atmospheric fine particles ($PM_{2.5}$) with high-resolution aerosol mass spectrometry and GC–MS, Environ. Sci. Technol., 45, 4854–4861, 2011.

---

## Author Comment (AC1) · 22 Aug 2017

We thank the referees for their comments, which helped improving the quality of our manuscript. A point by point response (in blue) to the reviewers' comments (in black, italics) will follow. Changes in the text are indicated in in black.

**Anonymous Referee #1**

*The manuscript presents the results obtained from the off-line analysis of filter samples collected at 9 sites in central Europe with different exposure characteristics. The study is mainly focused on the identification of the different sources that contribute to the organic aerosol loadings using PMF analysis. The obtained results indicate that biomass burning is a major contributor to primary organic aerosol with 88% in the alpine valley and 43% north of the alpine crest. On the contrary, the sum of HOA, COA and BBOA contributes less at the sites north of the alpine crest than at the southern alpine valley. Secondary organic aerosol production is enhanced during summer due to the increased biogenic emissions with temperature. Finally, it is estimated that primary biological particles which cannot be resolved by PMF could contribute significantly to PM10 organic aerosol.*

*The manuscript is very well written, coherent and easy to follow. A lot of effort has been put into the selection of the PMF solutions and the subsequent sensitivity analysis in order to provide the most sane and justified selection of factors, given the statistical nature of the analysis. This paper can be recommended for publication after some minor corrections listed below:*

*1) As the title reads "Long-term chemical analysis and organic aerosol source apportionment.." I would expect a short description of the trends in the chemical species as well of the 9 sites. It would be helpful as in a subsequent section the correlation with ammonium and nitrates is mentioned.*

We have mentioned in the manuscript (page 3, lines 11-14) that two papers regarding the offline analysis of this dataset are planned. This section has been modified as follows:

"… This paper focuses on the identification of the main factors influencing the OA concentrations at the different sites and the assessment of the associated uncertainties. In a second paper, we will investigate the site-to-site differences and general trends in the factor time series and their relationship with external parameters. …"

In the corrected version of the manuscript we have added a discussion on the fraction of ammonium that can be attributed to nitrate and to sulfate, for different seasons. This section reads as follows (P13 L30-33):

"… Here, we have used ammonium as a proxy for aged aerosols affected by anthropogenic emissions, as WOOA correlates better with ammonium than with nitrate sulfate. We note that in winter, when WOOA is highest, 56% of ammonium can be attributed to nitrate, whereas in summer ammonium sulfate dominates (97% of ammonium can be attributed to sulfate). Therefore, WOOA correlates more with nitrate ($R^2$ =0.64) than sulfate ($R^2$ =0.48). …"

*2) The possibility that WOOA could partially originate from the oxidation of BBOA could also be mentioned. During BB events ammonium is in a significant excess compared to sulfate, and this could possibly explain the good correlation between WOOA and $NH_4^+$.*

As we have mentioned above, the excess of ammonium compared to sulfate in winter is attributed to ammonium nitrate, which is thermodynamically more stable under lower temperatures and

higher relative humidity. As both WOOA and nitrate originate from a similar process – i.e. oxidation of precursors during winter time – both species do correlate. However, we do not think that this correlation is sufficient evidence to support that WOOA could partially originate from the oxidation of BBOA. We indeed think that this is the case based on modelling results we have recently published (Ciarelli et al., 2017) and on the chemical analysis of the same samples on a molecular level, using ultra-high mass resolution spectrometric techniques. The latter results will be presented in an upcoming publication.

*Minor corrections:*
*Abstract L21: … smaller than 10 μm from 9 stations …*

The text has been adapted to … smaller than 10 μm at 9 stations…

"… We present offline-AMS measurements for particulate matter smaller than 10 μm at 9 stations in central Europe with different exposure characteristics for the entire year of 2013 (819 samples). …"

*P16, L26: … closely correlates with NH4+.*

The text has been corrected accordingly.

"…WOOA, dominant SOA category during winter, closely correlates with $NH_4^+$. …"

**References:**

Allan, J. D., Jimenez, J. L., Williams, P. I.,Alfarra, M. R., Bower, K. N., Jayne, J. T., Coe, H., and Worsnop, D. R.: Quantitative sampling using an Aerodyne aerosol mass spectrometer 1. Techniques of data interpretation and error analysis, J. Geophys. Res., 108, 4090, doi:10.1029/2002JD002358, 2003.

Bozzetti, C., Daellenbach, K., R., Hueglin, C., Fermo, P., Sciare, J., Kasper-Giebl, A., Mazar, Y., Abbaszade, G., El Kazzi, M., Gonzalez, R., Shuster Meiseles, T., Flasch, M., Wolf, R., Křepelová, A., Canonaco, F., Schnelle-Kreis, J., Slowik, J. G., Zimmermann, R., Rudich, Y., Baltensperger, U., El Haddad, I., and Prévôt, A. S. H.: Size-resolved identification, characterization, and quantification of primary biological organic aerosol at a European rural site, Environ. Sci. Technol., 50, 3425-3434, doi:10.1021/acs.est.5b05960, 2016.

Bozzetti, C., Sosedova, Y., Xiao, M., Daellenbach, K. R., Ulevicius, V., Dudoitis, V., Mordas, G., Byčenkienė, S., Plauškaitė, K., Vlachou, A., Golly, B., Chazeau, B., Besombes, J.-L., Baltensperger, U., Jaffrezo, J.-L., Slowik, J. G., El Haddad, I., and Prévôt, A. S. H.: Argon offline-AMS source apportionment of organic aerosol over yearly cycles for an urban, rural and marine site in Northern Europe, Atmos. Chem. Phys. Discuss., doi:10.5194/acp-2016-413, 2017a.

Bozzetti, C., El Haddad, I., Salameh, D., Daellenbach, K. R., Fermo, P., Gonzalez, R., Minguillón, M. C., Iinuma, Y., Poulain, L., Müller, E., Slowik, J. G., Jaffrezo, J.-L., Baltensperger, U., Marchand, N., and Prévôt, A. S. H.: Organic aerosol source apportionment by offline-AMS over a full year in Marseille, Atmos. Chem. Phys., 17, 8247-8268, https://doi.org/10.5194/acp-17-8247-2017, 2017b.

Chen, Q, Miyazaki, Y., Kawamura, K., Matsumoto, K., Coburn, S. C., Volkamer, R., Iwamoto, Y., Kagami, S., Deng, Y., Ogawa, S., Ramasamy, S., Kato, S., Ida, A., Kajii, Y., and Mochida, M.: Characterization of chromophoric water-soluble organic matter in urban, forest, and marine aerosols by HR-ToF-AMS analysis and excitation emission matrix spectroscopy, Environ. Sci. Technol., 50, 10,351–10,360, 2016.

Corbin, J. C., Othman, A., Allan, J. D., Worsnop, D. R., Haskins, J. D., Sierau, B., Lohmann, U., and Mensah, A. A.: Peak-fitting and integration imprecision in the Aerodyne aerosol mass spectrometer: effects of mass accuracy on location-constrained fits, Atmos. Meas. Tech., 8, 4615-4636, doi:10.5194/amt-8-4615-2015, 2015.

Cubison, M. J. and Jimenez, J. L.: Statistical precision of the intensities retrieved from constrained fitting of overlapping peaks in high-resolution mass spectra, Atmos. Meas. Tech., 8, 2333–2345, doi:10.5194/amt-8-2333-2015, 2015.

Sun, Y., Zhang, Q., Zheng, M., Ding, X., Edgerton, E. S., and Wang, X.: Characterization and source apportionment of water-soluble organic matter in atmospheric fine particles (PM2:5) with high-resolution aerosol mass spectrometry and GC–MS, Environ. Sci. Technol., 45, 4854–4861, 2011.

---

## Author Comment (AC2) · 22 Aug 2017

We thank the referees for their comments, which helped improving the quality of our manuscript. A point by point response (in blue) to the reviewers' comments (in black, italics) will follow. Changes in the text are indicated in in black.

**P. Paatero (Referee)**

*This work develops first a mathematical machinery whereby they attempt to formulate a small number of factor spectra (2 assumed "known" mass spectra plus 4 fitted mass spectra) whereby a large number (819) of mass spectra, measured in 9 locations in different times of the year, could be sucessfully fitted. In the second part, they perform a very large number of repetitions of the modeling, so that different details of the model are varied. From variation of the results, they deduce reliability estimates for the obtained spectral profiles and contribution time series.*

*The mathematical operations are not adequately explained. It is impossible to know what they have done (and why) in different stages of the work. One of their key concepts is a constraining of F factor elements. Unfortunately, the defining equation (3) is so unclear (and possibly contains a typo) that I cannot even guess what they might mean by this equation. This review is limited to the first part of the work, modeling the measured mass spectra by a bilinear model. Although the second part is important, and has required a large amount of work, it cannot be analyzed in the time that would be available for such analysis. Thus I do not comment on the second part. I recommend that this manuscript should be published after the listed problems in mathematical presentation and elsewhere have been corrected.*

*Problems in the mathematical presentation*

*The fundamental principle of science is repeatability. Used methods should be defined clearly and in sufficient detail so that colleagues will be able (at least in principle) to repeat what was done by the authors. In the following, I discuss mathematics that have not been described in an understandable manner. In general, equations are the language of mathematics. Mathematical work should be described using equations. Verbal explanations may only help in understanding the equations, they cannot replace equations. In order to use equations, it is necessary to define symbols for various quantities. If necessary, use two-letter symbols.*

*Consider typographic questions: The prime should be avoided. Complicated notation in subscripts or in superscripts is often very difficult to read.*

We have taken into account the reviewer comments, clarified the mathematical operations used and their purpose and added/modified mathematical equations when needed. In the following we respond to the different points raised by the reviewer and indicate the changes in the text we have made.

*P5 L9-11 (page 5, lines 9-11)*
*.. by the estimated organic matter (OM) concentration, calculated as the product of the OC concentrations .. and the OM/OC ratios from the AMS measurements.*
*Reading this verbal explanation, it is very hard to understand what was done. In fact, I would need to try to write the equations in order to understand. Please do this equation writing for the benefit of your readers!*

We have added an additional equation (now equation 3), to clarify the quantification operation. The text reads as follows:

"…

Input data and error matrices consisted of 202 organic ions. The organic fragments, $x'_{i,j}$, obtained from offline AMS analyses do not directly represent ambient concentrations. Therefore, the signal of each fragment was converted to such an ambient concentration ($x_{i,j}$ in μg m$^{-3}$), by multiplying the fraction of this signal with the estimated organic matter (OM) concentration. The latter was calculated as the product of the OC concentrations measured by the Sunset OC/EC analyzer and the OM/OC ratios from the offline AMS measurements (OM/OC)$_{oAMS}$ (Eq. 3). Note that such scaling does not change the outcome of Eq. 2 since both data and error matrices are scaled in the same manner and the fingerprints ($f_{k,j}$) are not changed.

$$x_{i,j} = \frac{x'_{i,j}}{\sum_i x'_{i,j}} * OC * (OM/OC)_{oAMS} \tag{3}$$

…"

*P5 EQ(3): I do not understand eq(3) at all.*
*- what are f_kn and f'_kn (or is it f_kn'), what difference is there between them?*
*- is there a typo in the equation? As shown, the equation appears impossible.*
*This is a key detail in the manuscript as it describes constraints that perhaps have never been published in similar work. It must be described so that it is understandable.*

We do agree with the reviewer that the equation is misleading. The constraints were performed in the same way as in Canonaco et al. (2013) and the equation in the text was adapted to the one in the mentioned publication. The modified text reads as follows:

"… The PMF algorithm was solved using the multilinear engine-2 (ME-2, Paatero, 1999). Normalization of the PMF solution during the iterative minimization process is disabled as implemented in SoFi (Canonaco et al., 2013). ME-2 enables an efficient exploration of the solution space by a priori constraining the $f_{k,j}$ elements within a certain range defined by the scalar a ($0 \leq a \leq 1$) from a starting value $f_{k,j}'$, such that the modelled $f_{k,j}$ in the solution satisfy Eq. 4:

$$f_{k,j} = f'_{k,j} + a * f'_{k,j} \tag{4}$$

$f_{k,j}'$ is the starting value used as a priori knowledge from previous studies and $f_{k,j}$ is the resulting value in the solution. …"

*What was included in matrix X, to be modeled by PMF. It remains unclear what information was contained in X. Section 2 lists a number of variables whose concentrations were measured. Were all these included in X, or were some of these included, or none of them?*

*P5 L8-9 say: Input data and error matrices included 202 organic ions. Do you mean that input data consisted of 202 organic ions? The formulation you used may also be interpreted so that among other data, input data also included 202 organic ions.*

This sentence was indeed misleading. **X** consisted of 202 ions (variables) measured over n time points. The text has been adapted accordingly:

"… Input data and error matrices consisted of 202 organic ions. …."

*Also regarding matrix X:*
*Input data ... were rescaled by the estimated organic matter (OM) concentration ... In PMF, one is allowed to rescale data rows in any way, provided that error rows are also scaled in the same manner. Thus your scaling is OK. However, it would help the reader a lot if you state briefly why to scale, what advantage was achieved by scaling. Your scaling (which does not change profile spectra in any way) is useful for plotting figure 9. On the other hand, your scaling influences (improves or worsens) correlations between factor values and marker concentrations. Please explain and/or correct. Note that this scaling does not change the computations performed according to Eq(4) in any way.*

The scaling procedure does not influence in any way the PMF calculations. However, as the offline AMS signals do not directly represent ambient concentrations the factors retrieved should be scaled to the real ambient concentrations, before any correlation with external markers and data interpretation is made. Thus the scaling could indirectly affect the result by influencing the solution selection. The scaling could be achieved before or after running the PMF (as it was mentioned correctly by the reviewer the scaling does not change the computations performed according to Eq. 4). We have scaled the data before running PMF for computation reasons as in this way we do not need to scale the results obtained from every PMF solution repeatedly. The text related to this section has been adapted as follows:

"…

Input data and error matrices consisted of 202 organic ions. The organic fragments, $x'_{i,j}$, obtained from offline AMS analyses do not directly represent ambient concentrations. Therefore, the signal of each fragment was converted to such an ambient concentration ($x_{i,j}$ in µg m$^{-3}$), by multiplying the fraction of this signal with the estimated organic matter (OM) concentration. The latter was calculated as the product of the OC concentrations measured by the Sunset OC/EC analyzer and the OM/OC ratios from the offline AMS measurements (OM/OC)$_{oAMS}$ (Eq. 3). Note that such scaling does not change the outcome of Eq. 2 since both data and error matrices are scaled in the same manner and the fingerprints ($f_{k,j}$) are not changed.

$$x_{i,j} = \frac{x'_{i,j}}{\sum_i x'_{i,j}} * OC * (OM/OC)_{oAMS} \tag{3}$$

…"

*P5 L5-7 (input errors):*

> *The input errors ... include ... the uncertainty related to ion counting statistics and ion-to-ion signal variability at the detector.*
*I understand that you counted the ions. Then, ion-to-ion signal variability is \*not\* a source of uncertainty. If ion current is measured, instead of counting ions, then ion signal variablity \*is\* a source of error. Please correct or clarify.*

The AMS does not count ions, but rather an integrated signal (bit ns) that can be related to ion counts by means of a single ion calibration. Thus single ion signal variability is a source of uncertainty for PMF analysis, in addition to counting statistics. Both sources of errors are included in Allan et al. (2003, alpha and I, respectively).

Allan, J. D., Jimenez, J. L., Williams, P. I.,Alfarra, M. R., Bower, K. N., Jayne, J. T., Coe, H., and Worsnop, D. R.: Quantitative sampling using an Aerodyne aerosol mass spectrometer 1. Techniques of data interpretation and error analysis, J. Geophys. Res., 108, 4090, doi:10.1029/2002JD002358, 2003.

*P5 L20-24*
*describe a complicated method for filling knowledge gaps in the known/assumed fixed factor spectra ("reference profiles") of HOA and COA. I am not fully convinced about the performance of this method. The "natural" alternative method is to leave the unknown elements in reference profiles (profiles of HOA and COA) as ordinary factor elements, to be fitted by ME-2 together with all "normal" non-fixed F factor elements, as explained in detail below. Please include this remark in the corrected ms, so that future colleagues are encouraged to follow the safer and simpler method instead of your complicated method.*

We added the following statement to the manuscript, based on the reviewer's suggestions:

"… Alternatively, such missing ions can be also treated as ordinary factor elements, to be fitted by ME-2 with all other ordinary factor elements. …"

*Using "constrained factors" based on known profiles of HOA and COA. This topic was very difficult to understand at first. It was not clear what is "constrained" by what. Now I assume that you mean the following: In all PMF runs, two constant F factors were used, i.e. two rows of factor F were defined as a-priori fixed, so that the values of these constant factors were set equal to previously known mass spectra of HOA and COA. Is this what you mean? – Using constant or constrained factors is not familiar to PMF users, not at all. Such unusual methodology must be carefully explained so that all readers have the possibility to understand what you have done. In particular, you should explain that using fixed factors is not the same as using inequalities in order to constrain factors to lie between upper and lower limits, set very close to each other. Also, you should go into technical details here, because it is possible to implement constant factors in two different ways in ME-2. You should guide your readers to the optimal usage. It is possible to use "constant factors" that reside in a different matrix, which is clumsy. The alternative is to keep all F factors in the same matrix but define that the elements in two first (say) rows of F are "locked", not allowed to change during the fitting process. These elements are set equal to the known profile values before initiating the fit. If there are gaps in the knowledge of HOA and COA, then those unknown elements in "locked" F rows should simply not be locked at all, so that they may obtain their best possible values during the fit.*

For a better readability we adapted the text to:

"… $f_{k,j}'$ is the starting value used as a priori knowledge from previous studies and $f_{k,j}$ is the resulting value in the solution. In all PMF runs (unless mentioned otherwise), we used the high resolution mass spectra for HOA and COA (cooking OA) from Crippa et al. (2013b) as constraints, i.e. two rows of $f_{k,j}$ were set equal to the mass spectra of HOA and COA. Ions that were present in our datasets but not in the reference profiles for HOA and COA were inferred from published unit mass resolution (UMR) profiles (Ng et al., 2011 and Crippa et al., 2013c). For this purpose, the fraction of signal at a specific m/z in the UMR reference spectrum ($f_{UMR,m/z}$) was compared to the fraction of signal of all ions at this *m/z* in the HR reference spectrum ($f_{HR,m/z}$). The difference $f_{UMR,m/z} - f_{HR,m/z}$ was used as entries in $f_{k,j}'$ for such missing ions. For these ions, an *a*-value of unity was set. For the other factors, the factor elements were fitted by ME-2. Alternatively, such missing ions can be also treated as ordinary factor elements, to be fitted by ME-2 with all other ordinary factor elements.

…"

*Use of constant factors or constrained factors often causes so-called "normalization conflicts". How did you protect your bilinear model against normalization conflicts? This is another important detail that should be communicated to colleagues who might follow your example.*

We believe that the reviewer comment on normalization conflicts is referring to the normalization equations for the factor contribution present in the default ME-2 instruction file. If so, when calling ME-2 from SoFi, these normalization equations are disabled.

We added this technical aspect to the manuscript (P5 L18-23):

"… The Source Finder toolkit (SoFi v.4.9, Canonaco et al., 2013) for Igor Pro software package (Wavemetrics, Inc., Portland, OR, USA) was used to configure the PMF model and for post- analysis. The PMF algorithm was solved using the multilinear engine-2 (ME-2, Paatero, 1999). Normalization of the PMF solution during the iterative minimization process is disabled as implemented in SoFi (Canonaco et al., 2013). ME-2 enables an efficient exploration of the solution space by a priori constraining the $f_{k,j}$ elements within a certain range defined by the scalar a ($0 \leq a \leq 1$) from a starting value $f_{k,j}'$, such that the modelled $f_{k,j}$ in the solution satisfy Eq. 4: …"

*P7 L16-17*
*> For each of the four PMF datasets, 2420 PMF runs were performed for*
*> evaluating the sensitivity of the model to the chosen a-value and the seed.*

*This statement mentions sensitivity of the model to random seed. The random seed determines the pseudorandom initial values of PMF fit. In plain language, this statement says that there were local solutions so that depending on seed, PMF iteration converged to different local solutions. Presumably, these solutions had comparable Q values because otherwise, Q values would be used for selecting between solutions. Now these different solutions are somehow pooled together and their presence is otherwise ignored.*

*The presence of multiple solutions should be properly reported (e.g. how many of PMF runs had multiple solutions, how many different individual solutions per PMF run were obtained at most and on the average, are the solutions rotationally equivalent having identical residuals of fit, etc.) There are no fixed rules on what to do with multiple solutions. On one extreme, it has been suggested that scientists may at will pick the one solution they like most and ignore the others. At the other extreme,*

*PMF modeling of such data may be considered failing if there are several local solutions with comparable Q values.*

For each of the four PMF datasets, 2420 PMF runs were performed for evaluating the sensitivity of the model to the chosen *a*-value and the seed. The quality of each of the 2420 PMF runs was individually assessed using the criteria lined out in Sec. 3.6. We assessed the environmental interpretability based on the correlation of factor time series and markers of the respective source. All solutions fulfilling these criteria are accepted and considered plausible. Using these criteria 331 PMF runs were selected for $PMF_{block}$ (for $PMF_{zue,isol}$ 230, for $PMF_{zue,reps}$ 99, and for $PMF_{1filter/month}$ 269). This information was added to the text. We found that 80% of the accepted solutions have an a-value≤0.3 for HOA and an a-value≤0.5 for COA. The output HOA and COA factor profiles are therefore not significantly variable and very similar to the input profiles, indicating that similar solutions were selected. Furthermore, the yearly average factor concentrations of all selected $PMF_{block}$ solutions after $R_k$ correction are now shown for the case of Zurich as an illustration in Fig. S5. The distributions of each of the different factors do not show more than 1 distinct mode, indicating that we do not have several populations of solutions. This information was added to the supplementary. In a last step, all accepted solutions are pooled together. In Sec. 3.3, we added a remark linking to Sec. 3.6:

The modified Sec. 3.6 reads:

"… Monte Carlo simulations were performed and simulations for which res-OC$_i$ distributions were significantly different from 0 ($Q_{25}<0<Q_{75}$, details in SI) were discarded until 500 acceptable simulations were found. Thereby, 331 PMF runs were selected for $PMF_{block}$ (230 for $PMF_{zue,isol}$, 99 for $PMF_{zue,reps}$, and 269 for $PMF_{1filter/month}$). Median factor time series and recovery parameters from all retained simulations were then determined and the interquartile range (IQR) represents our best estimate of the uncertainties for the single PMF datasets. The Monte Carlo process was repeated for the four different PMF datasets described above and the resulting median time series of their estimated uncertainties were compared. The resulting uncertainty estimates and the method are described in Sec. 4.2.1. and in the SI. …"

The added information in the SI reads:

"…

Cumulative density functions for the a-values of HOA and COA are presented for the accepted solutions in Fig. S4. We found that 80% of the accepted solutions have an a-value≤0.3 for HOA and an a-value≤0.5 for COA. The output HOA and COA factor profiles are therefore not significantly variable and very similar to the input profiles, indicating that similar solutions were selected. Furthermore, the yearly average factor concentrations of all selected $PMF_{block}$ solutions after $R_k$ correction are shown for the case of Zurich as an illustration in Fig. S5. The distributions of each of the different factors do not show more than 1 distinct mode, indicating that we do not have several populations of solutions.

[Figure]

**Figure S4: Cumulative density functions of a-values for HOA and COA for the accepted solutions.**

The yearly average factor concentrations of all selected $PMF_{block}$ solutions after $R_k$ correction areshown for the case of Zurich as an illustration (Fig. S5). The distributions of each of the different factors do not show more than 1 distinct mode.

[Figure]

**Figure S5: Histograms of yearly average factor concentrations of all selected $PMF_{block}$ solutions (after $R_k$ correction)....”**

*If DISP is used for uncertainty estimation of a case with several local solutions, one often obtains the outcome that the model is "Not Well Defined" or "NWD". I would not suggest what the authors should do with their many-solution cases in addition to discussing them. Whatever they opt to do, they should describe it: what was done and why.*

The procedure that we have adopted aims at assessing the sensitivity of the PMF to the constrained profiles (a-value), to the pseudo-random starting point (seed), to the measurement uncertainties (by

including repeated measurements from one of the sites) and to the input data/rotational ambiguity (bootstrap). We have discarded a number of solutions based on a set of criteria. The PMF runs satisfying these criteria were all considered as plausible and discussed in the manuscript. Differences amongst these solutions are used as an estimate of the combined uncertainties mentioned above. The process is described in Sec. 4.2.1. and in the Supplementary Information. We added the following sentence at the end of Sec. 3.6 linking to Sec. 4.2.1.

"… The resulting uncertainty estimates and the method are described in Sec. 4.2.1. and in the SI. …"

*Section 3.3, sensitivity analysis*
*I cannot comment more on this analysis because I do not understand what the a-values are and how they were used.*

We do agree with the reviewer that the equation is misleading. The constraints were performed in the same way as in Canonaco et al. (2013) and the equation in the text was adapted to the one in the mentioned publication. The modified text reads as follows:

"…

The PMF algorithm was solved using the multilinear engine-2 (ME-2, Paatero, 1999). Normalization of the PMF solution during the iterative minimization process is disabled as implemented in SoFi (Canonaco et al., 2013). ME-2 enables an efficient exploration of the solution space by a priori constraining the $f_{k,j}$ elements within a certain range defined by the scalar a ($0 \leq a \leq 1$) from a starting value $f_{k,j}'$, such that the modelled $f_{k,j}$ in the solution satisfy Eq. 4:

$$f_{k,j} = f_{k,j}' + a * f_{k,j}' \tag{4}$$

$f_{k,j}'$ is the starting value used as a priori knowledge from previous studies and $f_{k,j}$ is the resulting value in the solution.
…"

*P6 L26-28 say:*
*> Paatero et al. (2014) compare the effectiveness in estimating modelling errors using two different approaches: the displacement (DISP) and bootstrap analysis (BS), respectively.*

*Here seems to be a terminological problem: in the quoted paper, Paatero et al. estimated the uncertainties of estimated F factor elements in the situation when no modelling errors are present. These F uncertainties depend mainly on random error in X and on rotational freedom of factor matrices G and F. It was specifically emphasized that the obtained uncertainties do not cover effects of modelling errors in the results. (Examples of modelling errors: non-constant factor profiles, wrong uncertainties assigned to data values.) Thus modelling errors were not estimated in the quoted paper.*

*P6 L29:*

*DISP involves running PMF several times using randomly perturbed factor profile elements of a reference solution*

*In fact, DISP estimation does not involve any randomness at all. F factor elements are pertubed in a systematic fashion by DISP. Perhaps here is confusion with Monte Carlo methods where random perturbations may be applied. DISP is not Monte Carlo.*

We apologize for the terminological mistake. The respective part of the manuscript has been corrected.

"… While this approach has been proven very effective in selecting a range of environmentally relevant solutions (Elser et al., 2016a, 2016b and Daellenbach et al., 2016), the resulting uncertainties may be underestimated. Paatero et al. (2014) compared the effectiveness in estimating uncertainties of factor elements using two different approaches: the displacement (DISP) and bootstrap analysis (BS). BS involves applying the model to input matrices consisting of a subset of the entire dataset. DISP involves running PMF several times using systematically perturbed factor profile elements of a reference solution, but allowing a defined difference in Q from the reference solution. Both approaches are computationally intensive, especially DISP. Because of such computational limitations the combination of BS and DISP was not feasible for the dataset presented here, especially in combination with *a*-value sensitivity tests. Therefore, we chose to perform 4 sensitivity tests performing PMF runs using 4 different input datasets, presented in the following. These sensitivity tests allow conclusions on the stability of PMF analysis when reducing the temporal or spatial resolution as well as the influence of the measurement repeatability.

..."

*In table (3), uncertainty estimates of percentage concentrations are not correctly computed.*

The overall uncertainties on the mass concentrations were estimated as the sum in quadrature of two error terms. The first relates to the variability in the yearly average mass concentrations for the different factors at the different sites calculated by changing the COA and HOA a-values using the $PMF_{block}$ setting (where all data points are considered, $\sigma_a$). The second error term relates to the additional variability in the yearly average mass concentrations for the different factors at the different sites due to the change in the input matrix ($PMF_{block}$, $PMF_{zue,isol}$, $PMF_{zue,reps}$). As the latter was determined only for the case of Zurich, then the same additional relative error ($\sigma_b$) was assumed for the other sites. This is indeed only a best estimate of the errors. In the corrected version of the manuscript we clarify the way these errors were determined (in Caption to Tab. 3) and we do not show errors on the percentage concentrations.

"…**Table 1: Yearly average contribution and uncertainty of resolved factors for $PMF_{block}$ run for the different sites and the average for all sites. The uncertainty is calculated based on the variability of the yearly averages from $PMF_{block}$ and the variability between the sensitivity tests. …**"

*Notation:*
*In supplement, subscript "i" is used as a subscript of Q. It is not defined what "i" means here. Does it mean number of factors? If yes, then the symbol used for number of factors should be used. If i does not have a definable meaning, then it might be clearer to omit the subscript in this case. In general, systematic use of subscripts would be a help for the reader. E.g. use i only as the index of sample (time), j as index of column of X, k index of factor. For other quantities, select other symbols and define what they mean.*

The index i in this figure and the text was removed.

*In different places, factor elements are denoted as G_ik and as g_ik. This may confuse readers. Eventually they will recognize that this difference does not mean anything, but first they will waste time trying to understand. Either, select one notation (preferred), or, in the section "Notation" (to be written), specify that G_ik and g_ik mean the same, and also F_kj and f_kj mean the same.*

$G_{ik}$ was replaced by $g_{ik}$.

*Eq (2) is incorrect. If you wish to use matrix element notation, then summation over k must be indicated. If you wish to use vector-matrix notation, then its use should be defined, especially because many of your readers are not familiar with such notation. In vector-matrix notation, index k would not be visible.*

Equation 2 and the adjacent text have been corrected:

"…

Source apportionment of the organic aerosol is performed using positive matrix factorization (PMF, Paatero, 1994). PMF is a statistical un-mixing model explaining the variability of the organic mass spectral data ($x_{i,j}$), as linear combinations of static factor profiles ($f_{j,k}$) and their time-dependent contributions ($g_{i,k}$), see Eq. 2 (where $p$ is the number of factors). The index $i$ represents a specific point in time, $j$ an ion, and $k$ a factor. The elements of the model residual matrix are termed $e_{i,j}$.

$$x_{i,j} = \sum_{k=1}^{p} g_{i,k} \, f_{k,j} + e_{i,j} \qquad\qquad (2)$$
…"

*P5 L10: sunset –> Sunset OC/EC analyzer*

Has been corrected.

*P5 L21*
*> Fitted ions in our datasets missing in the reference …*
*What do you mean by "fitted ions"? I do not understand this sentence.*

The sentence was rephrased to:

"…Ions that were present in our datasets but not in the reference profiles for HOA and COA were inferred from published unit mass resolution (UMR) profiles (Ng et al., 2011 and Crippa et al., 2013c). …"

*P7 L21:*
*> The identity of HOA and COA were identified first as their mass spectra were initially constrained.*

*Why do you need to identify HOA and COA? I would assume that they are on preselected rows of F, such as rows 1 and 2. No identification is needed for factors that are in known positions. What is wrong here? Am I understanding all this completely wrong?*

Yes, indeed HOA and COA are on preselected rows of F. The text has been adapted to:

"… Since HOA and COA were initially constrained on preselected rows of F, they did not need to be identified. …"

*Supplement*
*Figure SI.1*
*shows ratios of obtained Q vs. expected Qexp. How were Qexp computed? Did you take into account that downweighted columns of X contribute very little to Qexp? How many downweighted columns were present?*

The results displayed in FigS1 refer to the distributions of $Q_{i,j}$ (median and quartiles and not to $Q/Q_{exp}$. $Q_{i,j}$ is computed as:

$$Q_{i,j} = \left(\frac{e_{i,j}}{s_{i,j}}\right)^2$$

We corrected the text and axis labels accordingly and refer only to $Q_{i,j}$.

In PMF$_{block}$, only 7 of 202 ions were downweighted by a factor 3. Therefore, the bias in Q between using the non-downweighted or the downweighted $s_{i,j}$ is smaller than the variability in Q for the same set of solutions. Thus we do not think that this bias has an influence on the results, all the more since the absolute Q-values are not used in the solution selection.

*Obtained ratios Q/Qexp are of the order of 10 for 6 factors. This indicates that there are one or several significant modeling errors. It has been assumed among atmospheric scientists that a ratio of 4, say, would not be significant, that it could be caused byrandom variations from the expected Q. This assumption is totally wrong. Such ratios (>1.5, say) always have a cause that preferably should be understood in a project where careful mathematical analysis is attempted.*

*Possible causes of modeling errors are: underestimation of random errors in mass spectra, variation of factor profiles with time and between sites, systematic errors in preprocessing m/z spectra, and spurious sporadic local sources that cannot be modeled by PMF. An attempt should be made at understanding and discussing those errors, even if the effect to obtained results cannot be eliminated any more at this final stage. One useful diagnostic is to examine contributions to Q from different m/z values, from different times of day and days of week, and so on.*

Variation of factor profiles between sites:
We have explicitly represented the average $Q_{i,j}$ per site and per season to examine the performance of the model and reasons behind the observed $Q_{i,j}$ values. As we have already pointed out in the manuscript the model does get the data at sites in the north better than at sites in the South during winter. We did attribute this difference to the variability in the wood burning source profiles, here represented by only one factor. Indeed, we do agree that additional reasons may result in the high $Q_{i,j}$ observed. These are discussed below.

First, as pointed out above we do represent sources at all sites with a single profile and the examination of the site-to-site median $Q_{i,j}$ values do indicate a discrepancy in the model performance between sites in the north and south of the Alps. In comparison to PMF$_{block}$, the average $Q_{i,j}$ for Zurich is slightly reduced for the same number of factors when only including 1 site in PMF (PMF$_{zue,isol}$, PMF$_{zue,reps}$, average $Q_{i,j}$ 6). The difference between the site that is best explained and

the site that is least explained is also approximately 6 (Fig. SI.1). Therefore, we do indeed attribute a significant part of the unexplained variability to variation in source profiles at different sites.

Variation of factor profiles with season:
We note that these differences can also occur because of representing SOA at different seasons with only two factors: For example the difference in Q values between the best explained season (April-Mai-June) and the least well explained season (January-February-March) is approximately 4 (Fig. SI.1). A consistent difference between Q for week-days and weekends cannot be found (Fig. S2).

Underestimation of random errors in mass spectra
As mentioned in the manuscript, the samples from Zurich were measured at two instances, 5 months apart. When performing PMF on either batch separately (PMF$_{zue,isol}$ and PMF$_{zue,rep}$, respectively) their average $Q_{i,j}$ is comparable (Q=6). Therefore, there there does not seem to be a difference in the representation of random errors between these measurements.

Systematic errors in preprocessing m/z spectra
As we have mentioned in the manuscript the error matrix considers the detector counting statistics and the variation in the background. However, additional uncertainties can result from peak fitting of the high resolution AMS data (attribution of signal at a nominal mass to several ions). This procedure is affected by errors in the m/z calibration accuracy and precision and by the peak width. These errors are not currently taken into account, but may result in a significant underestimation of the measurement uncertainties especially for overlapping peaks, which may explain (at least partially) the high average $Q_{i,j}$.

In Fig. S3, we present the average $Q_{i,j}$ of individual ions. There is no clear dependence on the ion molecular weight and therefore, m/z. However, the average $Q_{i,j}$ for ions with a small mass defect (nominal mass – exact ion mass) is higher (~10) than for the other ions (~3). There are a lot of ions with small mass defects close to 0.03 a.m.u. in our dataset. This makes these peaks prone to overlap with other ions and thus their error subject to underestimation because this effect is not considered in the s$_{ij}$ calculation.

Sporadic local sources that cannot be modeled by PMF:
The ion families CHN and especially CS show higher average Q than CH, CHO1, and CHOgt1. Since CH$_3$SO$_2^+$ shows an event-driven time series, at least the high Q related to CS ions could be related to the inability of PMF to resolve these events accurately.

The corresponding section in the SI has been thoroughly reworked and reads now:

"…

Based on the input data for PMF$_{block}$, we evaluate the influence of the number of factors, p, on $Q_i$. For this experiment, both the traffic and cooking signatures were constrained using adapted reference profiles from Crippa et al. (2013b) as described in section III.1. Based on this evaluation, we chose to perform PMF using 6 factors.

$Q_{i,j}$ is computed using the PMF residuals (e$_{ij}$) and the PMF input errors (s$_{i,j}$):

$$Q_{i,j} = \left(\frac{e_{i,j}}{s_{i,j}}\right)^2 \tag{S1}$$

[Figure]

**Figure S1:** $Q_{i,j}$ **as a function of the number of factors for a reference experiment with all data used in PMF (9 sites, full year 2013, HOA and COA constrained with a=0.0 (b and d).** $\Delta(median(Q_{i,j}))_{max}$ **is evaluated for the different periods during the year 2013 (January-February-March, April-Mai-June, July-August-September, October-November-December) and for all sites (a and c). The grey line depicts the difference between the category (geographical or season) with the highest and the lowest median** $Q_{i,j}$**.**

Fig. S1 shows $Q_{i,j}$ s as a function of the number of factors for different sites (b) and seasons (d) and the difference between the highest (a) and lowest (c) median to evaluate the maximal difference in the mathematical quality of the solutions. As expected, forcing PMF to explain the variability in the dataset only with the 2 constrained factors ($p$=2), results in very high median $Q_{i,j}$ . $\Delta(median(Q_{i,j}))_{max}$ shows the difference in the median $Q_{i,j}$ between groups of points like sites or season. The smaller the $\Delta(median(Q_{i,j}))_{max}$, the smaller are the differences in the mathematical quality of the PMF solution for the different seasons/sites. To explain the temporal and geographical variability at least 5 factors are required. However, the difference between the site that is best explained and the site that is least explained is approximately 6 when using 5 or 6 factors. When increasing to 6 factors, also a factor explaining the variability of sulfur-containing organic ions (especially, $CH_3SO_2^+$) is resolved. Therefore, we opted to perform PMF using 6 factors. Using 6 factors, there is also no difference between the average $Q_{i,j}$ on week-days and weekend (Fig. S2).

[Figure]

**Figure S2: $Q_{i,j}$ as a function of the day of the week.**

However, for PMF$_{block}$ also with 6 factors, the average $Q_{i,j}$ is clearly larger (7 only the Zurich data points) than the ideal value of 1, i.e. the PMF residuals are larger than the measurement uncertainties. In comparison to PMF$_{block}$, the average $Q_{i,j}$ for Zurich is slightly reduced for the same number of factors when only including 1 site in PMF (PMF$_{zue,isol}$, PMF$_{zue,reps}$, average $Q_{i,j}$ 6). In this study, we analyse yearly cycles and, thereby, assume constant factor profiles throughout the year which can contribute to Q>1.

Another possible reason for Q>1 is an underestimation of the measurement uncertainty. A main contributor in high-resolution AMS data treatment (attribution of the signal at a nominal mass to several ions) stems from errors in the *m/z* calibration which could not be incorporated in the current data analysis. Recent studies demonstrate that for overlapping peaks (ions) the measurement uncertainties are strongly underestimated (Cubison et al., 2015; Corbin et al., 2015). For PMF$_{block}$ using 6 factors, average $Q_{i,j}$ do not depend on *m/z* but rather on the ion family (Fig. S3): ions consisting of C, H, S, (and O) summarized under the name (CS) and ions consisting of C, H, N, (and O) summarized under the name CHN have a higher $Q_{i,j}$ than hydrocarbon ions (CH, only C and H) and oxygenated ions (CHO$_{z=1}$ with 1 oxygen and CHO$_{z>1}$1 with more than 1 oxygen). Since the time series of CH$_3$SO$_2^+$ is event-driven, the high $Q_{i,j}$ of this ion hints to the fact that PMF is unable to accurately resolve all of these events.

The average $Q_{i,j}$ for ions with a mass defect (nominal mass – exact ion mass) around 0.03 a.m.u. is higher than for the other ions (Fig S3). Mass defects in this range are most common in our dataset. This makes these peaks prone to overlap with other ions and thus their error prone to an underestimation because this effect is not considered in the s$_{ij}$ calculation (described above).

[Figure]

**Figure S3: a)** Average $Q_{i,j}$ of ions in PMF$_{block}$ as a function of their mass-to-charge ratio (*m/z*). **The ions are color-coded with their composition (CH: ions consisting only of C and H; CHO1: ions consisting of C, H, and 1 O; CHOgt1: ions consisting of C, H, and more than 1 O; CHN: ions consisting of C, H, N, (and O); CS: ions consisting of C, H, S, (and O)). b)** Average $Q_{i,j}$ of the ions in **PMF$_{block}$ as a function of their mass defect (exact mass – nominal mass) as well as a histogram of the number of ions with a certain mass defect. The mean $Q_{i,j}$ of the ion families is displayed separately.**

Cumulative density functions for the a-values of HOA and COA are presented for the accepted solutions in Fig. S4. We found that 80% of the accepted solutions have an a-value≤0.3 for HOA and an a-value≤0.5 for COA. The output HOA and COA factor profiles are therefore not significantly variable and very similar to the input profiles, indicating that similar solutions were selected. Furthermore, the yearly average factor concentrations of all selected PMF$_{block}$ solutions after $R_k$ correction are shown for the case of Zurich as an illustration in Fig. S5. The distributions of each of

the different factors do not show more than 1 distinct mode, indicating that we do not have several populations of solutions.

[Figure]

**Figure S4: Cumulative density functions of a-values for HOA and COA for the accepted solutions.**

The yearly average factor concentrations of all selected PMF$_{block}$ solutions after $R_k$ correction areshown for the case of Zurich as an illustration (Fig. S5). The distributions of each of the different factors do not show more than 1 distinct mode.

[Figure]

**Figure S5: Histograms of yearly average factor concentrations of all selected PMF$_{block}$ solutions (after $R_k$ correction).**

*... "*

*Table SI.1*
*Why is there a table for mass closure criteria (used for rejecting bad solutions) when all entries in this table are identical? The criteria seem to concern distribution of residuals of OC fitting. Is it really so*

*that the fit is rejected if 1st quartile point is negative and 3$^{rd}$ quartile point is positive? In other words, the fit is rejected if residuals are symmetrical around zero. Usually, such residuals would be considered desirable.*

Symmetrical residuals around 0 were considered desirable and solutions were accepted if the first quartile was smaller than 0 and the third quartile larger than 0. The title of the Table was changed to:

"**Table S1: set of acceptance criteria used. r is the correlation coefficient between a factor time series and the respective marker. Q25 is the 1$^{st}$ quartile and Q75 the 3$^{rd}$ quartile.**"

**References:**

Allan, J. D., Jimenez, J. L., Williams, P. I.,Alfarra, M. R., Bower, K. N., Jayne, J. T., Coe, H., and Worsnop, D. R.: Quantitative sampling using an Aerodyne aerosol mass spectrometer 1. Techniques of data interpretation and error analysis, J. Geophys. Res., 108, 4090, doi:10.1029/2002JD002358, 2003.

Bozzetti, C., Daellenbach, K., R., Hueglin, C., Fermo, P., Sciare, J., Kasper-Giebl, A., Mazar, Y., Abbaszade, G., El Kazzi, M., Gonzalez, R., Shuster Meiseles, T., Flasch, M., Wolf, R., Křepelová, A., Canonaco, F., Schnelle-Kreis, J., Slowik, J. G., Zimmermann, R., Rudich, Y., Baltensperger, U., El Haddad, I., and Prévôt, A. S. H.: Size-resolved identification, characterization, and quantification of primary biological organic aerosol at a European rural site, Environ. Sci. Technol., 50, 3425-3434, doi:10.1021/acs.est.5b05960, 2016.

Bozzetti, C., Sosedova, Y., Xiao, M., Daellenbach, K. R., Ulevicius, V., Dudoitis, V., Mordas, G., Byčenkienė, S., Plauškaitė, K., Vlachou, A., Golly, B., Chazeau, B., Besombes, J.-L., Baltensperger, U., Jaffrezo, J.-L., Slowik, J. G., El Haddad, I., and Prévôt, A. S. H.: Argon offline-AMS source apportionment of organic aerosol over yearly cycles for an urban, rural and marine site in Northern Europe, Atmos. Chem. Phys. Discuss., doi:10.5194/acp-2016-413, 2017a.

Bozzetti, C., El Haddad, I., Salameh, D., Daellenbach, K. R., Fermo, P., Gonzalez, R., Minguillón, M. C., Iinuma, Y., Poulain, L., Müller, E., Slowik, J. G., Jaffrezo, J.-L., Baltensperger, U., Marchand, N., and Prévôt, A. S. H.: Organic aerosol source apportionment by offline-AMS over a full year in Marseille, Atmos. Chem. Phys. Discuss., doi:10.5194/acp-2017-54, in review, 2017b.

Chen, Q, Miyazaki, Y., Kawamura, K., Matsumoto, K., Coburn, S. C., Volkamer, R., Iwamoto, Y., Kagami, S., Deng, Y., Ogawa, S., Ramasamy, S., Kato, S., Ida, A., Kajii, Y., and Mochida, M.: Characterization of chromophoric water-soluble organic matter in urban, forest, and marine aerosols by HR-ToF-AMS analysis and excitation emission matrix spectroscopy, Environ. Sci. Technol., 50, 10,351–10,360, 2016.

Corbin, J. C., Othman, A., Allan, J. D., Worsnop, D. R., Haskins, J. D., Sierau, B., Lohmann, U., and Mensah, A. A.: Peak-fitting and integration imprecision in the Aerodyne aerosol mass spectrometer: effects of mass accuracy on location-constrained fits, Atmos. Meas. Tech., 8, 4615-4636, doi:10.5194/amt-8-4615-2015, 2015.

Cubison, M. J. and Jimenez, J. L.: Statistical precision of the intensities retrieved from constrained fitting of overlapping peaks in high-resolution mass spectra, Atmos. Meas. Tech., 8, 2333–2345, doi:10.5194/amt-8-2333-2015, 2015.

Sun, Y., Zhang, Q., Zheng, M., Ding, X., Edgerton, E. S., and Wang, X.: Characterization and source apportionment of water-soluble organic matter in atmospheric fine particles (PM2:5) with high-resolution aerosol mass spectrometry and GC–MS, Environ. Sci. Technol., 45, 4854–4861, 2011.

---

## Author Comment (AC3) · 22 Aug 2017

We thank the referees for their comments, which helped improving the quality of our manuscript. A point by point response (in blue) to the reviewers' comments (in black, italics) will follow. Changes in the text are indicated in in black.

**Anonymous Referee #3**

*Comments on "Long-term chemical analysis and organic aerosol source apportionment at 9 sites in Central Europe: Source identification and uncertainty assessment" by Daellenbach et al. The manuscript presents new research which clearly fits within the scope of the journal. The text is well-written and fairly easy to follow. Some of figures, however, compile several information and are not as straightforward to interpret (e.g. Figure 8) – please make sure to modify them (color axis, split into subplots, etc.) to improve readability.*

We removed the second axis (PBOA) in Figure 8 since the difference is only a scaling factor. We improved the readability of several figures.

*The technique described here is a follow-up of the characterization of OA measurements based on filter collections followed by water extraction and analysis by HR-AMS, previously published, being the novelty a large statistics from 9 sampling sites and, most importantly, PMF analysis of the OA spectrum from filters. Although the former is unquestionably of scientific interest, it is the latter that will allow others to apply the technique and indeed reach its goals as described in the introduction. At its current stage, the manuscript doesn't fully achieve it.*

Major comments:
*\* The description depth of the PMF applied to this very specific dataset doesn't seem to be proportional to its level of development in regard to the widely used techniques. Please detail it more carefully.*

Based on the comments of reviewer 2 and 3 we have significantly adapted the parts related to the description of the PMF. We also highlighted more clearly the part of the model that is typically applied and the other parts that are developed within this work. For example, PMF is widely used, but the ME-2 implementation is not. Also, in the revised version of the manuscript, we do describe more clearly the steps we have adopted for the PMF solution selection.

"…

[revised manuscript text omitted]

*…"*

*\* Section 4.2.2 seems quite weak, three methods to estimate PBOA are presented, but no clear conclusion is given other than it underestimates based on previous literature. From my perspective this section doesn't add too much to the manuscript and could easily be removed, however if the authors wish to keep it, please make sure to better constrain the methods into a valid scientific output.*

We believe that removing this section would be misleading and not sufficiently transparent and would hide the fact that SOOA might be overestimated due to the contribution of PBOA, which could not be separated by PMF. Our main objective is not quantifying PBOA, but rather its potential contribution to SOOA. Therefore, with the three approaches presented we attempt to estimate this PBOA contribution. These approaches suggest that PBOA would contribute between 0.3 and 1.0 give units, or 18-58% of SOOA during the warm season. In the revised version of the manuscript, we have clarified the aim of the section related to PBOA contribution estimation. This section reads as follows:

"… Unresolved sources in PMF are an inherent uncertainty of source apportionment analyses. As Bozzetti et al. (2016) show, PBOA can present considerable contributions to OA in PM10 (constituting a large part of coarse OA). In the present analysis, PBOA could not be separated by PMF (neither unconstrained nor using the mass spectral signature from Bozzetti et al., 2016). This inability might be caused by the low water-solubility and the absence of PM2.5 filters in the dataset. Since these coarse particles are only abundant in PM10 and not in PM2.5/PM1, the presence of both PM10 and PM2.5 samples, exhibiting a large gradient in PBOA, might allow an unambiguous separation of PBOA. The aim of this section is to estimate the influence of PBOA on the source apportionment results. A quantification of this fraction is, however, beyond the scope of this paper. In the following, we estimate the influence of PBOA in three alternative ways: …

*…"*

*Minor comments:*
*\* Abstract. L.21: add the word "from" between μm and 9.*

As suggested by another reviewer, we added "at".

*\* Abstract. L.24: remove "which is" and add a comma before related.*

The text has been corrected.

*\* P.2, L.31: Please remove "restricted to WSOA in".*

The respective part has been removed.

*\* P.12, L.10: Remove the word "here".*

The word "here" was removed.

*\* External gas-phase tracers (besides the use of NOx just to separate HOA, COA) could also add some information of the surrounding chemistry – for example, what is the ozone (over 24h, or just afternoon) in regard to SOOA and WOOA? And Ox?*

For Zurich, we added a comparison of SOOA concentrations to ozone and Ox ($O_3$+$NO_2$):

"...

In Figure S7, we compare the SOOA concentrations to ozone and Ox ($O_3$+$NO_2$) for Zurich. The SOOA concentrations follow best the temperature ($R_{s,SOOA,temp}$=0.65, Fig. S7.a) but show also some correlation to ozone $R_{s,SOOA,O3}$=0.33, Fig. S7.b) and Ox ($R_{s,SOOA,Ox}$=0.38, Fig. S7.c).

[Figure]

[Figure]

[Figure]

**Figure S7: SOOA concentrations compared to temperature, ozone, and Ox ($O_3$+$NO_2$) for Zurich.**

..."

---

## Author Comment (AC4) · 22 Aug 2017

We thank the referees for their comments, which helped improving the quality of our manuscript. A point by point response (in blue) to the reviewers' comments (in black, italics) will follow. Changes in the text are indicated in in black.

**Anonymous Referee #4**

**General comments:**

*In this paper the concentrations of the six types of organic aerosol (OA) components (HOA, COA, BBOA, WOOA, SOOA, and SC-OA) over Switzerland are reported based on the off-line analysis of the water-soluble aerosol components in aerosol samples using an aerosol mass spectrometer (AMS). The characteristics of the retrieved OA components, e.g., the relative abundances and seasonality, are presented. Further, the uncertainty of the concentrations of the retrieved OA is discussed. The source identification of OA components based on long-term samplings at multiple locations is important, and the application of the aerosol mass spectrometry for the chemical analysis of aerosol samples collected on filters made it possible in this study. The contributions of the major sources of OA to the atmospheric concentrations in the studied area have been characterized well in view of location and seasonality. Although the results presented in this paper are highly valuable, this paper needs substantial improvement in terms of the presentation quality. The explanations for the statistical analyses are not fully comprehensive, and a part of them would be flawed. Further, the point of this study is not very clear because both the methodology of the analysis itself and the results based on it are presented and discussed. To make the point clearer, it may be better to move the discussion on the uncertainty based on the results in Figures 6 and 7 to the experimental section or the supplement. Other minor issues regarding the presentation quality include inadequate explanations, undefined abbreviations/symbols, and grammatical errors. For the reasons above, substantial improvement is required for the publication of this paper in its final form. More specific comments are listed below.*

**Specific comments:**
*Page 3, 1st paragraph: It may be better to explain more about previous source apportionment studies for organic aerosols using off-line AMS measurement techniques. The group of the first and corresponding authors reported two more studies, both of which were also for European sites (Bozzetti et al., 2017a, 2017b). There are also other source apportionment studies based on statistical analysis for the mass spectra obtained using off-line AMS techniques (Sun et al., 2011; Chen et al., 2016). Emphasis should be on which characteristics of atmospheric aerosols have not been studied tentatively even by the use the off-line AMS techniques.*

In the revised version of the manuscript, we have mentioned previous work that used a similar methodology to here. While Chen et al. (2016) have used factor analysis for the mass spectra obtained using off-line AMS techniques; their focus was on the identification of chromophores. Indeed, the studies by Bozzetti et al. (2017a, b) from our group use the same methodology for a similar purpose that is the determination of spatially resolved trends of OA sources. However, the sites studied and the challenges faced in the aforementioned studies are different from here. Here, as the sites are not homogeneous the main aim of the paper is providing a methodology to satisfactorily represent the OA by few factors, with a systematic and objective assessment of the results and the underlying uncertainties.

This paragraph reads now:

"… This approach allows the retroactive investigation of specific events, e.g. haze events in China (Huang et al., 2014) as well as AMS measurements of coarse mode aerosol (Bozzetti et al., 2016) and long-term source apportionment studies (Bozzetti, 2017b, 2017a). Such an approach was also used in recent studies for identifying the different types of water-soluble chromophores (Chen et al., 2016). Additionally, such filters are routinely collected and are already available over multi-year periods at many air quality monitoring stations around the world for years/decades. …"

*Page 3, 2nd paragraph: The chemical analysis using the AMS was limited to the watersoluble component of organics in PM10, although the water-insoluble organic component was also taken into consideration in the source apportionment. This point should be addressed more explicitly.*

Indeed, the chemical analysis using the AMS was limited to the water soluble fraction. However, using factor-specific recoveries determined in Daellenbach et al. (2016), the contribution of the different factors to WSOC could be scaled to OC. This is described in detail in Section 2.5.

*Page 3, line 13: The site-to-site differences and time series are not explained in a specific part of this paper.*

This sentence was misleading. Detailed analysis of the site-to-site differences and time series will be presented in a second paper. The sentence has been adapted to:

"…In a second paper, we will investigate the site-to-site differences and general trends in the factor time series and their relationship with external parameters. …"

*Page 4, lines 1-3: How were the mass spectra of the extracts from aerosol samples corrected for field blanks? Because the sensitivity of an AMS to aerosol components depends on the particle size, the signal intensity of organics should not be proportional to the organic mass flux from the nebulizer. For this reason, the assessment of the blank level is not straightforward. More explanation to this point is necessary.*

Between two samples we measured ultrapure water. The recorded signal was subtracted from the sample spectra. In a previous study, we showed that the organic blank measurements collected by ultrapure water nebulization provide a comparable blank estimate to the organic blanks determined from the nebulization of $NH_4NO_3$ (Bozzetti et al., 2017a). However, we also analyzed field blanks which were extracted and measured in the same way as the exposed samples.
…
We added accordingly a statement in the manuscript on P3 L30 –P4 L7:

"… The measurement blank was determined before and after every filter sample. Each sample was recorded for 480 seconds (AMS V-mode, *m/z* 12-447), with a collection time for each spectrum of 30 seconds. Ultrapure water was measured for 720 seconds. Once per day, ultrapure milliQ water was nebulized with a particle filter interposed between the nebulizer and the AMS, for the determination of the gas-phase contribution to the measured mass spectrum, which was then subtracted during analysis from both blanks and filter samples. The filters from Zurich were analysed twice with a time difference of approximately 5 months to assess the measurement repeatability. High resolution mass spectral analysis was performed for each *m/z* (mass to charge) in the range of 12- 115. The measurement blank was subtracted from the sample spectra. In a previous study, it has been shown that the measurement blank is comparable to the organic blanks obtained from the nebulization of $NH_4NO_3$ (Bozzetti et al. (2017a). The interference of $NH_4NO_3$ on the $CO_2^+$ signal described by Pieber et al. (2016) was corrected as follows (Eq. 1): …"

We have previously shown that the organic signal from the nebulization of MQ water is not statistically significantly different from the organic signals from the nebulization of $NH_4NO_3$ (Bozzetti et al. (2017a), which might potentially act as a carrier seed of contaminants. Therefore, we considered the MQ water to be an adequate representation of the measurement background. In addition to the measurement blanks, we have measured field blanks following the same procedure. These samples showed WSOC and OC concentrations higher than instruments detection limits. As this contamination can contribute to different extents to different factors, data have been corrected post PMF as described in Section 2.5.

In order to account for the effect of the field blanks on the source apportionment, we subtracted the blank concentrations factor after the PMF analysis. To that purpose we performed PMF runs using $PMF_{block}$ while also the field blank measurements were included in the PMF run. Thereby, we found how much the different factors contributed to the field blanks. Finally, we subtracted this effect from the factor time series. In addition, the OC blank levels used in the previous version of the manuscript were overestimated and have now been updated. Therefore, all numbers in the manuscript related to the source apportionment analysis slightly changed (yearly average factor concentrations changed by around 15%). However, the main conclusions remain the same.

The respective text was adapted for a better readability:

*"… For a limited number of PMF runs ($PMF_{block}$) also the field blank analyses were included in the PMF input data. This provides the contribution of different factors to the field blanks which were used to correct the output factor time series. Uncertainties induced by the blank subtraction were propagated. …"*

*Page 4, lines 9-10: The expression in the parenthesis is unclear and needs to be reworded.*

The text has been adapted in the revised version of the manuscript:

*"… The correction factor $\left(\frac{CO_{2,meas}}{NO_{3,meas}}\right)_{NH_4NO_3,pure}$ was determined based on measurements of aqueous $NH_4NO_3$ conducted regularly during the entire measurement period and varied between ~1% and ~5% (Pieber et al., 2016). …"*

*Page 5, lines 9-11: The method for rescaling here and that explained in the 2nd paragraph of page 9 does not seem identical.*

While OC concentrations are available for all samples, WSOC concentrations are only available for a subset of all samples (Magadino and Zurich). Therefore, for the samples from Magadino and Zurich it was possible to evaluate mass closure, i.e. whether the sum of WSOC factor concentrations after Rk correction matched the measured OC ($OC_{i,res} = OC_{i,meas} - \sum_k WSOC_{i,k}/R_k$. For all other samples this was not possible because of the unavailability of WSOC concentrations and, therefore, these samples needed to be scaled to OC.

The text has been adapted in the revised manuscript:

"… The last criterion relates to OC mass closure. A Monte Carlo approach was applied to evaluate whether a combination of water soluble factor time series and recovery parameters would achieve OC mass closure, as described in the following. For the samples from Zurich and Magadino, for which WSOC concentrations were available (in contrast to the other samples), offline AMS measurements were scaled to the water soluble organic matter (WSOM), calculated using the WSOC measurements and OM/OC ratios from the AMS HR analysis. The water-soluble contributions from an identified aerosol source in a sample $i$ were rescaled to their total organic matter concentrations ($OA_{i,k}$), where k represents a given factor, using combinations of factor recoveries as determined by Daellenbach et al. (2016, medians of the used combinations being: $R_{HOA}$: 0.11, $R_{COA}$: 0.54, $R_{BBOA}$: 0.65, and $R_{OOA}$: 0.89 used for WOOA and SOOA). …

*Page 5, equation 3: The constraint represented by equation 3 seems erroneous because the left and the right parts of the equation are identical.*

There was indeed a typo in the equation. The equation was adapted to the presentation in Canonaco et al. (2013), since the interface presented therein was used and the same type of constraints was applied.

*Page 5, lines 21-22: Were the inferred fitted ions also for constraint? Does this sentence mean all the factors other than HOA and COA were inferred from published UMR profiles?*

Ions that were present in our dataset but not in the reference profiles for HOA and COA were inferred and constrained. However, such ions were given an a-value of unity. For the other factors besides HOA and COA, factor elements were not constrained but fitted by ME-2. The section was adapted to:

"… $f_{k,j}'$ is the starting value used as a priori knowledge from previous studies and $f_{k,j}$ is the resulting value in the solution. In all PMF runs (unless mentioned otherwise), we used the high resolution mass spectra for HOA and COA (cooking OA) from Crippa et al. (2013b) as constraints, i.e. two rows of $f_{k,j}$ were set equal to the mass spectra of HOA and COA. Ions that were present in our datasets but not in the reference profiles for HOA and COA were inferred from published unit mass resolution (UMR) profiles (Ng et al., 2011 and Crippa et al., 2013c). For this purpose, the fraction of signal at a specific m/z in the UMR reference spectrum ($f_{UMR,m/z}$) was compared to the fraction of signal of all ions at this $m/z$ in the HR reference spectrum ($f_{HR,m/z}$). The difference $f_{UMR,m/z} - f_{HR,m/z}$ was used as entries in $f_{k,j}'$ for such missing ions. For these ions, an *a*-value of unity was set. For the other factors, the factor elements were fitted by ME-2. Alternatively, such missing ions can be also treated as ordinary factor elements, to be fitted by ME-2 with all other ordinary factor elements.…"

*Page 5, lines 22-24: The explanation in this sentence is not clear. This sentence should be reworded.*

The respective sentence has been reworded to:

" … For this purpose, the fraction of signal at a specific m/z in the UMR reference spectrum ($f_{UMR,m/z}$) was compared to the fraction of signal of all ions at this $m/z$ in the HR reference spectrum ($f_{HR,m/z}$). The difference $f_{UMR,m/z} - f_{HR,m/z}$ was used as entries in $f_{k,j}'$ for such missing ions …"

*Page 8, line 1: The values of the recoveries used in this study should be presented.*

Recoveries are presented in Fig. 6 and the text was adapted to:

The water-soluble contributions from an identified aerosol source in a sample *i* were rescaled to its total organic matter concentration ($OA_{i,k}$), where *k* represents a given factor, using combinations of factor recoveries as determined by Daellenbach et al. (2016, medians of the combinations being: $R_{HOA}$: 0.11, $R_{COA}$: 0.54, $R_{BBOA}$: 0.65, and $R_{OOA}$: 0.89 used for WOOA and SOOA).

*Page 8, line 2: The meaning of "the contributions of different factors to the field blank samples" is not clear. What was done is not clear, either.*

The text was adapted for a better readability:

"… For a limited number of PMF runs (PMF_block) also the field blank analyses were included in the PMF input data. This provides the contributions of different factors to the field blanks which were used to correct the output factor time series. Uncertainties induced by the blank subtraction were propagated. …"

*Page 8, line 27: Is "α_=0.5" the significance level? Fifty percent is too high.*

For PMF_block, we performed an additional sensitivity test with α=0.05 instead of α=0.5. The results are described in a new section in the supplementary material and the comparison mentioned in the main text.

The sentence in the main text reads:

"… A t-test is then used to verify the significance (α=0.5) of the average correlation coefficient between factor and marker time series, r_avg (Eq. 7):

$$t_{avg} = \frac{r_{avg}}{\sqrt{\frac{1-r_{avg}^2}{N-2}}}$$

(7)

Here, $r_{avg}$ is the correlation coefficient averaged over the different stations, derived from the average z value, $t_{avg}$ is the corresponding t-value and N is the average number of samples at the different stations. Results with significance level α=0.05 are summarized in Fig. S8.

…"

The section in the supplementary information reads:

"...

For PMF$_{block}$, a sensitivity test with significance level of 0.05 instead of 0.5 as in the base case was performed. The factor concentrations and their corresponding uncertainties ($\sigma_a$) are compared and displayed as number density functions (Fig. S8). Changes in the estimated factor concentrations are within 10% of the factor concentrations for SCOA and smaller for all other factors. The uncertainty related to COA is decreased when lowering the significance level to 0.05, while the other factors remain largely unaffected.

[Figure]

**Figure S8: number density functions of source apportionment results obtained using a significance level of 0.05 normalized to results obtained using a significance level of 0.5: a) Comparison of factor concentrations b) Comparison of uncertainty estimate ($\sigma_a$).**

..."

*Page 8, line 26-28: How the statistical analysis using the average values from different stations can be justified? The validity of this method is not obvious.*

The ratios of factor concentrations to marker concentrations cannot be assumed to be the same at all sites. Therefore, correlation coefficients need to be calculated at different sites. However, to achieve an optimized system for the entire dataset the average *R* should be considered.

*Page 9, lines 2-4: More details in the calculation should be given so that the readers can assess its validity.*

We have adapted the text and added further information:

"...

The first two criteria (1-2) ensure an appropriate separation of HOA and COA from OOA and BBOA, respectively. Criteria 3-6 relate to the evaluation of the correlation between factor and marker time

series. This was achieved by computing the Fisher-transformed correlation coefficient z at different stations (Eq. 6):

$$\mathbf{z = 0.5 * ln\left(\frac{1+r}{1-r}\right) = arctan(r)} \tag{6}$$

where r is the correlation coefficient between factor and marker at a given station. The obtained z values at the different stations are subsequently averaged and transformed back to $r_{avg}$ before further analysis. A t-test is then used to verify the significance (α=0.5) of the average correlation coefficient between factor and marker time series, $r_{avg}$ (Eq. 7):

$$\boldsymbol{t_{avg} = \frac{r_{avg}}{\sqrt{\frac{1-r_{avg}^2}{N-2}}}} \tag{7}$$

Here, $r_{avg}$ is the correlation coefficient averaged over the different stations, derived from the average z value, $t_{avg}$ is the corresponding t-value and N is the average number of samples at the different stations. Results with significance level α=0.05 are summarized in Fig. S8.

To evaluate whether HOA correlated significantly better with NOx than COA did, the average z values obtained between HOA and NOx and between COA and NOx (Eq. 6) were compared, using a standard error on the z distribution of $1/\sqrt{N-3}$ (Zar, 1999).

…"

*Page 9, lines 14-16: Is the issue really explained in the supplement?*

*This paragraph has been reworded and further information has been added. Now the paragraph reads:*

"…The sum of $OC_{i,k}$ from all factors *k* (mod-$OC_i$) was then evaluated against the measured OC (meas-$OC_i$). For this, the residual OC mass (res-$OC_i$) for each sample was calculated (meas-$OC_i$ – mod-$OC_i$), and the residual distributions were examined for different conditions that are specified in the Supplement. In summary, a solution was only accepted if res-$OC_i$ were normally distributed around 0 considering all points and subsets of points: a) summer, b) winter c) Magadino, d) Zurich, e) low and high concentrations of the single factors (see Table S1). …"

*Page 10, line 2: What are the percentages of the accepted data?*

We added this information to the manuscript at P10 L5:

"…Thereby Thereby, 331 PMF runs were selected for $PMF_{block}$ (230 for $PMF_{zue,isol}$, 99 for $PMF_{zue,reps}$, and 269 for $PMF_{1filter/month}$). …"

*Page 11, line33 – page 12, line 1: This sentence is not clear. Does COA relate to the discussion here?*

Besides the correlation between the yearly average concentrations of SC-OA and NOx, we also present the same values for HOA vs. NOx and COA vs. NOx. This comparison allows the conclusion that not all anthropogenically influenced factors show a relation to NOx.

"… SC-OA instead exhibits low background levels episodically intercepted by remarkable ten-fold enhancements, especially at urban sites affected by traffic emissions (e.g. the SC-OA contribution is significantly higher at sites with higher yearly $NO_X$ average levels). The hypothesis of an influence of traffic activity on SC-OA is provided by the correlation of the yearly average concentrations with NOx ($R_{s,SC-OA,NOx}$ =0.65, n=9, p<0.06) which is, however, comparable to the correlation of HOA and COA (e.g., $R_{s,HOA,NOx}$=0.68, n=9, p<0.05, $R_{s,COA,NOx}$ =0.68, n=9, p<0.05).. …"

*Page 13, line 1: The "uncertainties" here should be relative uncertainties. This should be addressed explicitly.*

In Figure 7, we present the uncertainties relative to the factor concentrations. We corrected the mistake in the manuscript:

"… We note that relative uncertainties related with SOOA increase with decreasing concentrations (Fig. 7). A small error in modelling sources with high contributions (BBOA, WOOA) in winter can result in a large error of SOOA with its small contribution during winter. Furthermore, some other sources like primary biological OA (PBOA, see Sec. 4.2.2) might also mix into SOOA. …"

*Page 13, lines 2-3: The meaning of "contribution from other more significant wintertime sources" is not clear. Further, justification of the explanation in this sentence should be provided.*

This sentence has been adapted and reads now:

"… A small error in modelling sources with high contributions (BBOA, WOOA) in winter can result in a large error of SOOA with its small contribution during winter. …"

*Page 13, lines 3-4: It is not clear why the mixing of some winter-time SOA into SOOA results in a larger uncertainty.*

SOOA concentrations are relatively small in winter compared to BBOA or WOOA concentrations. Therefore, a small error in modelling BBOA or WOOA can result in a rather big error in SOOA. The corresponding section in the manuscript was adapted:

"… We note that relative uncertainties related with SOOA increase with decreasing concentrations (Fig. 7). A small error in modelling sources with high contributions (BBOA, WOOA) in winter can result in a large error of SOOA with its small contribution during winter. Furthermore, some other sources like primary biological OA (PBOA, see Sec. 4.2.2) might also mix into SOOA. …"

*Page 14, lines 1-2: How was σb calculated?*

12 samples are present in all PMF datasets (PMFblock, PMFzue,iso, PMF1filt/month, PMFzue,reps). For these 12 samples we determine the median concentration for the independently treated PMF datasets. $\sigma_b$ is the variability of the median concentrations of these 12 samples. This information can be found in the supplementary information (section uncertainty estimation and propagation).

We adapted the main text to the following:

"… The variability of the factor time series for the single PMF sensitivity tests (PMF$_{block}$, PMF$_{zue,isol}$, PMF$_{1filter/month}$, PMF$_{zue,eps}$) is used as an uncertainty estimate (shaded area in Fig. 4). This estimate ($\sigma_a$) depends on the measurement repeatability (10 single mass spectra included for each sample) and on the selected PMF solution/ $R_k$ combinations and, therefore, also on the *a*-value. However, the variability depending (1) on the choice of input points (time and site; PMF$_{block}$, PMF$_{zue,isol}$, PMF$_{1filter/month}$) and (2) on the instrumental reproducibility (PMF$_{zue,reps}$) of the offline AMS measurements is not accounted for. The contribution of (1) and (2) to the uncertainty is assessed through the sensitivity tests by examining the variability of the median factor time-series ($\sigma_b$). $\sigma_b$ is the variability of the median factor concentrations from the PMF sensitivity tests using PMF$_{block}$, PMF$_{zue,isol}$, PMF$_{1filter/month}$, PMF$_{zue,reps}$ for the 12 samples common to all 4 PMF datasets. For the 12 filters common in all PMF datasets (PMF$_{block}$, PMF$_{zue,isol}$, PMF$_{1filter/month}$, PMF$_{zue,reps}$), we calculate a best estimate of the overall uncertainty (err$_{tot}$), by propagating both error terms: $\sigma_a$ and $\sigma_b$. …"

*Page 14, line 8: The meaning of "σb – rotational ambiguity" is not clear.*

The part "- rotational ambiguity" has been removed and more information has been added to the text:

"… It is worthwhile to note that for major factors exhibiting a similar seasonality, i.e. WOOA and BBOA, a great part of the uncertainty arises from $\sigma_b$. Thus the variability between the PMF solutions using PMF$_{block}$, PMF$_{zue,isol}$, PMF$_{1filter/month}$, PMF$_{zue,reps}$ ($\sigma_b$ ) and, therefore, the sensitivity of the factor concentrations on the chosen PMF dataset significantly contribute to the uncertainty. …"

*Page 14, lines 12-15: This sentence is not very organized and needs to be reworded.*

We reworded the sentence, now it reads:

"… In the present analysis, PBOA could not be separated by PMF (neither unconstrained nor using the mass spectral signature from Bozzetti et al., 2016). This inability might be caused by the low water-solubility and the absence of PM2.5 filters in the dataset.  …"

*Page 14, lines 24-26: What is the definition of the site-to-site variability? Was standard deviation calculated for the average values at respective sites?*

The site-to-site variability is the standard deviation of average concentrations at the different sites. This information has been added to the text and the sentence has been reworded for easier readability:

"…Using this approach, we estimate that PBOA contributes 0.30 µg/m$^3$ during the warm months (site-to-site variability computed as standard deviation of the average concentration of all sites of 0.03 µg/m$^3$). During the same period, SOOA concentrations are 1.78 µg/m$^3$ (site-to-site variability of 0.18 µg/m$^3$) and OA concentrations 4.32 µg/m$^3$ (site-to-site variability of 0.44 µg/m$^3$).  …"

*Page 15, line 2: The use of the word "however" does not seem appropriate.*

The word "however" has been removed and the sentence reworded. Now it reads:

"…The ion $C_2H_5O_2^+$ (indicator for PBOA) shows higher concentrations with increasing $OC_{coarse}$ concentrations. …"

*Page 16, line 5: Is POA here the sum of HOA, COA and BBOA? Shouldn't it be defined here instead of line 9?*

We moved the definition of POA to line 9. Now the paragraph reads:

"…In general, the seasonality of the factor time series is consistent for all the 9 sites in the entire study area (Fig. 9). In summer, SOOA is the main contributor to OA, while in winter POA (HOA+COA+BBOA) becomes more important although WOOA still contributes significantly. In comparison to the sites in northern Switzerland, OA in the southern alpine valleys is dominated by BBOA in winter, while in the north WOOA also plays a role. The different factors contribute 0.47±0.12 (HOA, average and site-to-site variability), 0.31±0.13 (COA), 1.37±1.77 (BBOA), 0.67±0.31 (SC-OA), 1.11±0.23 (WOOA), 1.31±0.13 (SOOA) $\mu g/m^3$ for all sites during the entire year (Table 3). In northern Switzerland, POA contributes less to OA (POA/OA=0.3) than in the southern alpine valleys where POA/OA is equal to 0.6. …"

*Figure 2: The aHOA and aCOA are not defined explicitly.*

The figure caption was adapted to:

"…**Figure 1: Step-by-step outline of adopted source apportionment approach (factor recoveries $R_k$). $a_{HOA}$ and $a_{COA}$ represent the a-value applied for HOA and COA, respectively. …**"

*Figure 3: The definition of fm/z should be given.*

The definition of fm/z has been added to the figure caption:
"…**Figure 2: PMF factor profiles of HOA, COA, BBOA, SOOA, WOOA, SC-OA, color-coded with ion family of PMF_block (average). fm/z is the relative intensity at a specific mass-to-charge ratio (m/z). …**"

*Figure 9: The definition of OAexpl is not given explicitly.*

The figure caption has been adapted to:

"…**Figure 3: Map of Switzerland with yearly cycles. Negative concentrations were set to 0 prior to normalization for display. The OA mass explained by the source apportionment analysis is termed $OA_{expl}$. …**"

*Page 2 (supplement): The relationship among "Qi/Qi;exp", "Δ (Qi/Qi;exp)", "Δ Qi/Qi;exp", "(Qi/Qi;exp contribution)", and "Δ (Qi/Qi;exp contribution)" is not clear.*

The nomenclature has been unified. The results refer to the distributions of $Q_{i,j}$ –(median and quartiles or average) and not to $Q/Q_{exp}$. $Q_{i,j}$ (referred to as Q-contribution) is computed as:

$$Q_{i,j} = \left(\frac{e_{i,j}}{s_{i,j}}\right)^2$$

We corrected the text and axis labels accordingly and refer only to Q-contribution.

*"… Δ(Q-contribution) shows the difference in the median Q-contributions between groups of points like sites or season. The smaller the Δ(Q-contribution), the smaller are the differences in the mathematical quality of the PMF solution for the different seasons/sites. …"*

*Page 3 (supplement): The definitions of "r(. . . )", "Q25(OCres)" and "Q75(OCres)" are not given.*

The table caption has been modified as:

**"…Table S1: set of acceptance criteria used. r is the correlation coefficient between a factor time series and the respective marker. Q25 is the 1ˢᵗ quartile and Q75 the 3ʳᵈ quartile. …"**

*Page 4 (supplement): The definition of "fion" is not given.*

The figure caption has been updated:

**"…Figure S6: mass spectral fingerprints of BBOA (PMF$_{block}$) and nebulized levoglucosan. $f_{ion}$ is the fraction of signal of a respective ion to the sum of the total signal."**

**Technical corrections:**
*Page 1, line 21: Should "at" be added between "10 _m" and "9 stations"?*

This mistake has been corrected.

*Page 3, line 20: "HiVol" should be spelled out.*

Spelled out as High-Volume samplers.

*Page 5, lines 29-30: The subscripts of "PMF" are not written consistently in the paper.*

The PMF datasets are now consistently called:
PMF$_{block}$
PMF$_{zue,isol}$
PMF$_{1filter/month}$
PMF$_{zue,reps}$

*Page 10, line 18: Should "from" be added after "profile"?*

The missing word has been added as well as the number corrected:

"…COA profile elements were constrained using the COA profile from Crippa et al. (2013b) and the obtained factor profile maintains the same features (OM/OC of 1.32, IQR 1.30-1.33, Fig. 3). …"

*Page 13, line 29: Should "Fig. 5" be "Fig. 4"?*

Yes it should be Fig. 4. The mistake has been corrected.

*Page 14, lines 30 and 33: Should "is in summer" be "in summer is"?*

The mistake has been corrected.

*Page 14, line 33: "OCcoarse" should be defined in line 30..*

OCcoarse is now defined on line 30.

"… Bozzetti et al. (2016) showed that coarse OC ($OC_{coarse} = OC_{PM10}-OC_{PM2.5}$) in summer is dominated by PBOA for samples collected at a rural site in Switzerland (Payerne). …"

*Page 19, lines 9-10: The list of authors are incorrect.*

The citation has been corrected:

"Daellenbach, K. R., Bozzetti, C., Křepelová, A., Canonaco, F., Wolf, R., Zotter, P., Fermo, P., Crippa, M., Slowik, J. G., Sosedova, Y., Zhang, Y., Huang, R.-J., Poulain, L., Szidat, S., Baltensperger, U., El Haddad, I., and Prévôt, A. S. H.: Characterization and source apportionment of organic aerosol using offline aerosol mass spectrometry, Atmos. Meas. Tech., 9, 23-39, doi:10.5194/amt-9-23-2016, 2016."

*Table 1: The commas after "St. Gallen" and "San Vittore" in the column "Site (station code)", and the periods after "m" in the column "altitude" should be omitted. The initial letter of "altitude" should be capitalized.*

In the column further information on the station location have been added. Altitude has been capitalized and the periods are omitted.

| Site (station code) | Classification | General location | Altitude |
|---|---|---|---|
| Basel, St. Johann (bas) | Urban/background | North of Alps/Swiss plateau | 308 m |
| Bern, Bollwerk (ber) | Urban/traffic | North of Alps/Swiss plateau | 506 m |
| Frauenfeld, Bahnhofstr. (fra) | Suburban/background | North of Alps/Swiss plateau | 403 m |
| Payerne (pay) | Rural/background | North of Alps/Swiss plateau | 539 m |
| St. Gallen, Rorschacherstr. (gal) | Urban/traffic | North of Alps/Swiss plateau | 457 m |
| Zurich, Kaserne (zue) | Urban/background | North of Alps/Swiss plateau | 457 m |
| Vaduz, Austrasse (vad) | Urban/traffic | North of Alps/alpine valley | 706 m |
| Magadino, Cadenazzo (mag) | Rural/background | South of Alps | 254 m |
| San Vittore, Zentrum (vi) | Rural/traffic | South of Alps/alpine valley | 330 m |

*Figure 4 caption: It is better to write "HOA, COA,. . . ." in the order of the corresponding panels.*

The figure caption has been adapted and reads now:

"…Figure 4**:** HOA, COA, BBOA, WOOA, SOOA, and SC-OA and respective marker concentrations as a function of time for Zurich in 2013. Depicted are the median factor time series results for the different PMF datasets (median) including the uncertainties for $PMF_{block}$ (first and third quartile) (green: $PMF_{block}$, black: $PMF_{zue,isol}$, red: $PMF_{zue,reps}$, pink bullets: $PMF_{1filter/month}$). … "

*Figure 5 caption: Should "NH4" be "NH4+"?*

The figure caption has been corrected:

"…Figure 5: Scatter-plots for the different extreme sensitivity tests for Zurich and for all sites for $PMF_{block}$ median concentrations): a) HOA vs $NO_x$, b) BBOA vs levoglucosan, c) SOOA vs temperature, d) WOOA vs $NH_4^+$. …"

*Figure 7: Should "[" after "concentration" be "]"?*

In the y-axis |concentration| refers to the absolute concentration.

*Page 4 (supplement): The "interquartile range PMF block" should be represented by a symbol because it is in a mathematical formula. It may be better to write "median bootstrap solutions" as the subscript of σ.*

Since $\sigma_a$ and $\sigma_b$ are explained in detail in the text we remove this part of the mathematical expression. The equation reads now:

„…

$$\text{err}_{i,k,\text{tot}} = \sqrt{\sigma_a^2 + \sigma_b^2}$$

…„

**References:**

Allan, J. D., Jimenez, J. L., Williams, P. I.,Alfarra, M. R., Bower, K. N., Jayne, J. T., Coe, H., and Worsnop, D. R.: Quantitative sampling using an Aerodyne aerosol mass spectrometer 1. Techniques of data interpretation and error analysis, J. Geophys. Res., 108, 4090, doi:10.1029/2002JD002358, 2003.

Bozzetti, C., Daellenbach, K., R., Hueglin, C., Fermo, P., Sciare, J., Kasper-Giebl, A., Mazar, Y., Abbaszade, G., El Kazzi, M., Gonzalez, R., Shuster Meiseles, T., Flasch, M., Wolf, R., Křepelová, A., Canonaco, F., Schnelle-Kreis, J., Slowik, J. G., Zimmermann, R., Rudich, Y., Baltensperger, U., El Haddad, I., and Prévôt, A. S. H.: Size-resolved identification, characterization, and quantification of primary biological organic aerosol at a European rural site, Environ. Sci. Technol., 50, 3425-3434, doi:10.1021/acs.est.5b05960, 2016.

Bozzetti, C., Sosedova, Y., Xiao, M., Daellenbach, K. R., Ulevicius, V., Dudoitis, V., Mordas, G., Byčenkienė, S., Plauškaitė, K., Vlachou, A., Golly, B., Chazeau, B., Besombes, J.-L., Baltensperger, U., Jaffrezo, J.-L., Slowik, J. G., El Haddad, I., and Prévôt, A. S. H.: Argon offline-AMS source apportionment of organic aerosol over yearly cycles for an urban, rural and marine site in Northern Europe, Atmos. Chem. Phys. Discuss., doi:10.5194/acp-2016-413, 2017a.

Bozzetti, C., El Haddad, I., Salameh, D., Daellenbach, K. R., Fermo, P., Gonzalez, R., Minguillón, M. C., Iinuma, Y., Poulain, L., Müller, E., Slowik, J. G., Jaffrezo, J.-L., Baltensperger, U., Marchand, N., and Prévôt, A. S. H.: Organic aerosol source apportionment by offline-AMS over a full year in Marseille, Atmos. Chem. Phys., 17, 8247-8268, https://doi.org/10.5194/acp-17-8247-2017, 2017b.

Chen, Q, Miyazaki, Y., Kawamura, K., Matsumoto, K., Coburn, S. C., Volkamer, R., Iwamoto, Y., Kagami, S., Deng, Y., Ogawa, S., Ramasamy, S., Kato, S., Ida, A., Kajii, Y., and Mochida, M.: Characterization of chromophoric water-soluble organic matter in urban, forest, and marine aerosols by HR-ToF-AMS analysis and excitation emission matrix spectroscopy, Environ. Sci. Technol., 50, 10,351–10,360, 2016.

Corbin, J. C., Othman, A., Allan, J. D., Worsnop, D. R., Haskins, J. D., Sierau, B., Lohmann, U., and Mensah, A. A.: Peak-fitting and integration imprecision in the Aerodyne aerosol mass spectrometer: effects of mass accuracy on location-constrained fits, Atmos. Meas. Tech., 8, 4615-4636, doi:10.5194/amt-8-4615-2015, 2015.

Cubison, M. J. and Jimenez, J. L.: Statistical precision of the intensities retrieved from constrained fitting of overlapping peaks in high-resolution mass spectra, Atmos. Meas. Tech., 8, 2333–2345, doi:10.5194/amt-8-2333-2015, 2015.

Sun, Y., Zhang, Q., Zheng, M., Ding, X., Edgerton, E. S., and Wang, X.: Characterization and source apportionment of water-soluble organic matter in atmospheric fine particles (PM2:5) with high-resolution aerosol mass spectrometry and GC–MS, Environ. Sci. Technol., 45, 4854–4861, 2011.